



# Brief Communication: The reliability of gas extraction techniques for analysing $CH_4$ and $N_2O$ compositions in gas trapped in permafrost ice-wedges

**Ji-Woong Yang[1*], Jinho Ahn[1], Go Iwahana[2], Sangyoung Han[1], Kyungmin Kim[1**] and Alexander Fedorov[3,4]**

[1]School of Earth and Environmental Sciences, Seoul National University, Seoul, South Korea

[2]International Arctic Research Center, University of Alaska, Fairbanks, USA

[3]Melinkov Permafrost Institute, Russian Academy of Science, Yakutsk, Russia

[4]North-Eastern Federal University, Yakutsk, Russia

 *Now at: Laboratoire des Sciences du Climat et de l'Environnement, LSCE/IPSL, CEA-CNRS-UVSQ, Université Paris-Saclay, Gif-sur-Yvette, France

**Now at: Division of Earth and Planetary Materials Science, Department of Earth Science, Graduate School of Science, Tohoku University, Sendai, Japan

**Correspondence**: Jinho Ahn (jinhoahn@snu.ac.kr)

**Abstract.** Methane ($CH_4$) and nitrous oxide ($N_2O$) compositions in ground ice may provide information on their production mechanisms in permafrost. However, existing gas extraction methods has not been well tested. We test conventional wet and dry gas extraction methods using ice-wedges from Alaska and Siberia. We find that both methods extract gas from the easily extractable parts of the ice (e.g., gas bubbles), and yield similar results for $CH_4$ and $N_2O$ mixing ratios. We also find insignificant effects of microbial activity during wet extraction. However, both techniques are unable to fully extract gas from the ice, presumably because gas molecules adsorbed onto or enclosed in soil aggregates are not easily extractable. Estimation of gas production in subfreezing environment of permafrost should consider the incomplete gas extraction.

## 1. Introduction

Permafrost soils preserve large amounts of soil carbon and nitrogen in a frozen state



(e.g., Hugelius et al., 2014; Salmon et al., 2018), removing this frozen carbon (C) and nitrogen
(N) from active global cycles. Therefore, future projections of permafrost stability are of great
interest, particularly because thawing permafrost may lead to decomposition and/or
remineralization of the buried soil C and N and their abrupt emission into the atmosphere in
the form of greenhouse gases (GHGs) – carbon dioxide ($CO_2$), methane ($CH_4$), and nitrous
oxide ($N_2O$), which in turn trigger positive feedbacks (e.g., Salmon et al., 2018). In addition,
the projected polar amplification (e.g., Masson-Delmotte et al., 2013) may strengthen these
positive feedbacks. However, the processes responsible for in-situ C and N remineralization
and GHG production in ground ice are poorly understood, despite the fact that ground ice
accounts for a substantial portion of the upper permafrost: up to approximately 40–90% by
volume of ice-rich permafrost, or Yedoma (e.g., Kanevskiy et al., 2013; Jorgenson et al., 2015).

41       The gases trapped in ground ice allow unique insights into the origin of ground ice and

evidence for in-situ microbial aerobic respiration (e.g., Lacelle et al., 2011). Among others, the
GHGs in ground ice may provide detailed information on in-situ biogeochemical processes
responsible for GHG production (e.g., Boereboom et al., 2013; Kim et al., 2019). However,
analytical methods remain poorly scrutinized. Boereboom et al. (2013) utilized the
conventional melting-refreezing method (wet extraction) used in polar ice core analyses. In this
technique, the ice samples were melted under a vacuum to liberate the enclosed gases, then
refrozen to expel the dissolved gases present in the meltwater. Other studies conducted by
Russian scientists used an on-site melting method in which a large (1–3 kg) block of ground
ice sample was melted in a saturated sodium chloride (NaCl) solution, in order to minimize
microbial activity and gas dissolution (Cherbunina et al., 2018 and references therein). A recent
study instead used a dry extraction technique to prevent the microbial activity during wet
extraction (Kim et al., 2019), which employed a needle-crusher in a vacuum to crush
approximately 10 g of ice sample without melting (Shin, 2014).



In this study, for the first time we test the reliability of both wet and dry extraction
methods for $CH_4$ and $N_2O$ mixing ratios and contents (volume or moles of gas in a unit mass
at standard temperature and pressure conditions (STP)) using permafrost ground ice samples.
Ice-wedge samples from Alaskan and Siberian permafrost were used because the ice-wedge is
one of the most abundant morphological features of massive ground ice, consisting of
approximately 5 to 50% by volume of the upper permafrost (Kanevskiy et al., 2013; Jorgenson
et al., 2015). More specifically, this study aims to address the following scientific questions: 1)
Do wet and dry extraction methods yield different results? 2) Are the melting-refreezing results
affected by microbial activity during gas extraction? 3) How effectively does the wet/dry
extraction extract gases from ice wedges? To address the first question, $CH_4$ and $N_2O$ results
from dry and wet extractions were compared. For the second question, we applied the wet
extraction method to both biocide-treated and control samples. Finally, for the third question
we carried out tests with and without extended number of hitting ice with a needle system in a
crushing chamber, as well as additional dry extraction from ice samples that had been degassed
by our wet extraction method.

**2. Materials and Methods**
**2.1.Ice samples and sample preparation**
The ice-wedge samples used in this study were collected from Churapcha, Cyuie
(central Yakutia), and Zyryanka (north-eastern Yakutia) in Siberia, as well as from northern
Alaska. The Churapcha site (61.97°N, 132.61°E) is located approximately 180 km east of
Yakutsk. The Cyuie site (61.73°N, 130.42°E) is located approximately 30 km southeast of
Yakutsk. The Cyuie samples were collected from two outcrops (CYB and CYC) (Kim et al.,
2019). At each site, 30 cm long ice-wedge cores were drilled perpendicular to the outcrop
surface.





Zyryanka is located in the southern boreal region of the Kolyma River, at the junction
of the Chersky and Yukaghir Ranges, in a region affected by thermokarst development
(Fedorov et al., 1991). Site A (Zy-A) is located on a tributary of the Kolyma River,
approximately 22 km north of Zyryanka. Site B (Zy-B) is approximately 14 km west of the
start of the Kolyma tributary, which begins ~11 km north of Zyryanka. Site F (Zy-F) is located
approximately 4 km west of the tributary that leads to site B. The ground ice samples were
collected from riverbank walls exposed by lateral erosion using a chainsaw. Most of the
outcrops that were sampled for ground ice were on the first terrace of the river.
For the Alaskan sampling locations, Bluff03 (69.40°N, 150.95°W) and Bluff06
(69.14°N, 150.61°W) are located in the Alaska North Slope region, approximately 120 and 150
km from the Arctic Ocean, or 100 and 70 km northwest of the Toolik Field Station (68.63°N,
149.59°W), respectively. Samples from Bluff03 were collected from the bluff walls that had
developed by gully formations on a gentle slope of the Yedoma using a chainsaw. Samples of
Bluff06 were collected from outcrops within eroded frozen peatland in a thaw lake basin. All
the ice-wedge samples used in this study were stored in a chest freezer at < -18°C before
analysis.
The ice-wedge ice is most different from polar ice cores, in that their gas mixing ratios
are not homogeneous (e.g., Kim et al., 2019), which may hinder exact comparison with results
from adjacent ice samples. We therefore randomly mixed sub-samples to reduce the effect of
the heterogeneous gas composition distribution (random cube method hereafter).
Approximately 100–200 g of an ice-wedge sample was cut into 25 to 50 cubes of 3–4 g each,
and for each experiment, ~10 to 12 cubes were randomly chosen so that the total weight of the
sub-sample was ~40 g.

**2.2. Gas extraction procedures**



### *Dry extraction (needle crusher)*

For dry extraction, we used a needle-crusher system at the Seoul National University (SNU, Seoul, South Korea) (Shin, 2014). In brief, 8~13 g of ice sample was crushed in a cold vacuum chamber (extraction chamber). The ice samples were usually hit five times by the needle set. The temperature within the extraction chamber was maintained at -37°C by using a cold ethanol-circulating chiller. The extracted gas was dried by passing it through a water vapor trap at -85°C and cryogenically trapping it in a stainless-steel tube (sample tube) at approximately -257 °C using a helium closed-cycle refrigerator (He-CCR). Since the extraction chamber cannot accommodate ~40 g of ice at once, the ~40 g of random cube sub-samples were extracted using three sequential extractions and the gas liberated from each extraction was trapped in a sample tube.

Following extraction, the sample tubes were detached from the He-CCR, warmed to room temperature (~20°C), and attached to a gas chromatograph (GC) equipped with an electron capture detector (ECD) and a flame ionization detector (FID) to determine the mixing ratios of $CH_4$ and $N_2O$. Details of the GC system are given in Ryu et al. (2018). The daily calibration curves were established using working standards of $15.6 \pm 0.2$ ppm $CH_4$, $10000 \pm 30$ ppm $CH_4$, $2960 \pm 89$ ppb $N_2O$, $29600 \pm 888$ ppb $N_2O$, and a modern air sample from a surface firn at Styx Glacier, Antarctica, which was calibrated as $1758.6 \pm 0.6$ ppb $CH_4$ and $324.7 \pm 0.3$ ppb $N_2O$ by the National Oceanic and Atmospheric Administration (NOAA).

### *Wet extraction (melt-refreeze)*

For the control and $HgCl_2$-treated wet extraction experiments, a melting-refreezing wet extraction system at SNU was employed (Yang et al., 2017; Ryu et al., 2018). The gas extraction procedure is identical to the procedure described in Yang et al. (2017) and Ryu et al. (2018), except for the sample gas trapping procedure (see below). Ice-wedge sub-samples of



~40 g (composed of 10–12 ice cubes for each) were placed in a glass container welded to a
stainless-steel flange (sample flask), and the laboratory air inside the sample flasks was
evacuated for 40 min. The sample flasks were then submerged in a warm (~50°C) tap water
bath to melt the ice samples. After melting was complete, the meltwater was refrozen by
chilling the sample flasks with cold ethanol (below -70°C). The sample gas in the headspace
of each sample flask was then expanded to the volume-calibrated vacuum line to estimate the
volume of extracted gas, and trapped in a stainless-steel sample tube by the He-CCR device.
In this study, we attached the He-CCR device to our wet extraction line and the gas samples in
the flasks were cryogenically trapped. The reasons for using He-CCR instead of direct
expansion to a GC are twofold: 1) to better compare the dry and wet extraction methods by
applying the same trapping procedure, and 2) to maximize the amount of sample gas for GC
analysis, because the gas expansion from a large flask allows only a small fraction of gas to be
measured by the GC.
For biocide-treated tests, 1.84 mmol of mercuric chloride ($HgCl_2$) was applied per unit
kilogram of soil, following established procedures for soil sterilization (Fletcher and Kaufman,
1980). Taking the dry soil mass of the analysed samples (0.33 g) into account, we added 24 µL
of saturated $HgCl_2$ solution (at 20°C) to the sample flasks. The flasks with $HgCl_2$ solution were
then frozen in a deep freezer at < -45°C to prevent the dissolution of ambient air into the
solution during ice sample loading. After the wet extraction procedure was complete, the
extracted gas was trapped in a sample tube and the $CH_4$ and $N_2O$ mixing ratios were determined
using the same GC-ECD-FID system as the dry-extracted gas. The resulting $CH_4$ and $N_2O$
mixing ratios have not been corrected for partial dissolution in ice melt in the flasks, because
$CH_4$ and $N_2O$ trapped in refrozen ice are negligible compared to the ranges of the systematic
blanks (see Appendix).



### 2.3.Gas content


The analytical methods described previously are for determining the mixing ratios of
$CH_4$ and $N_2O$ in the extracted gas. To convert these mixing ratios into moles of $CH_4$ and $N_2O$
per unit mass of ice-wedge sample ($CH_4$ and $N_2O$ content, respectively, hereafter) requires data
regarding the amount of gas extracted. The gas content is a measure of gas volume enclosed in
a unit mass of ice sample at STP (in mL $kg_{ice}^{-1}$). Thus, the $CH_4$ and $N_2O$ contents can be
calculated using the gas content, the total mass of the random cube ice, and the gas mixing
ratio. The gas content in the control and $HgCl_2$-treated wet extraction experiments was
calculated from the temperature and pressure of the extracted gas and the internal volume of
the vacuum line. The details of the extraction system and correction methods used for
estimating gas content are described in Yang (2019). Similarly, the gas content of the dry
extraction samples was also inferred from the volume and pressure of gas inside the vacuum
line once the sample tube was attached to the line for GC analysis. The uncertainties of the
calculated $CH_4$ and $N_2O$ contents were calculated by using error propagation of the blanks and
gas content uncertainties (see Appendix for uncertainty estimation of the blank corrections and
gas contents).

### 2.4.Dry soil content


Dry soil content was measured using the leftover meltwater from the control-wet
extraction tests. After the control-wet extractions were complete, the sample flasks were shaken
well and the meltwater samples were each poured into a 50 mL conical tube. The meltwater
and soils were separated by a centrifugal separator at 3000 rpm for 10 min. The separated wet
soils were wind-dried in evaporating dishes at approximately 100°C for 24 hours. The weight
of each individual evaporating dish was pre-measured before use. The dry soil content was
calculated by subtracting the weight of the evaporating dish from the total weight of the dried



soil sample plus the evaporating dish.

**3.  Results and Discussion**
**3.1.Comparison between wet and dry extraction methods**
The results from the wet and dry extractions were compared using 23 ice-wedge
samples (21 for $N_2O$) from Alaska and Siberia. In both the $CH_4$ and $N_2O$ mixing ratio analyses,
we found that the wet and dry extraction results did not differ significantly ($p > 0.1$), regardless
of sampling site or soil content (Figure 1, a to d). We noted that the heterogeneous distribution
of gas mixing ratios may not have been completely smoothed out by our sub-sample selection,
although we randomly chose 8–12 ice cubes for each measurement. Some previous studies
have avoided using the wet extraction method because of potential reactivation of microbial
$CH_4$ and/or $N_2O$ production in ice melt (e.g., Cherbunina et al., 2018; Kim et al., 2019).
Assuming that activation of microbial metabolism is unlikely during dry extraction at a
temperature of -37°C in the extraction chamber for < 1 h, our findings may imply that wet
extraction does not stimulate microbial reactivation to a measurable extent.



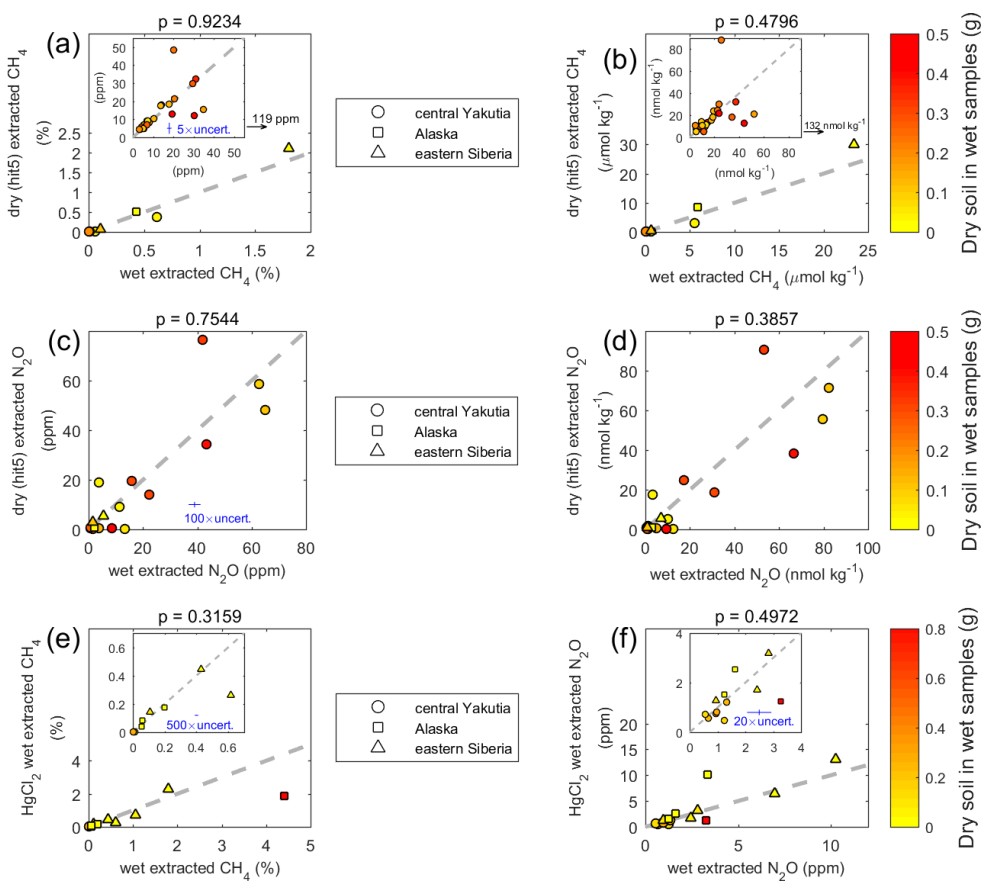

**Figure 1.** Comparison of $CH_4$ and $N_2O$ mixing ratios and contents obtained by different extraction methods. Shown are scatter plots between wet- and dry (hit5) extraction results of $CH_4$ (a and b) and $N_2O$ (c and d), and between control- and biocide-treated wet extraction results for $CH_4$ (e) and $N_2O$ (f). Left panels (a, c, and e) and (f) present in mixing ratios of gas in bubbles, while right (b) and (d) panels in moles of gas in a unit mass of ice (gas content). The sampling locations are indicated by different symbols. The color of each data point indicates the dry soil weight in the subsamples used in control wet extraction. The 1-sigma uncertainties of the mixing ratios (a, c, e, and f) are denoted as blue error bars (see Appendix). The error bars are not visible where the error bars are smaller than markers. The grey dashed lines are 1:1 reference line. Note that the units of the axes of the insets in (e) and (f) are identical to the original plots. The p-value of two-sided Students' t-test of each comparison is denoted at the top of each plot.



### 3.2. Testing microbial alteration during wet extraction


To test the microbial production of $CH_4$ and $N_2O$ during wet extraction more accurately,
we conducted wet extraction experiments on samples treated with $HgCl_2$, a commonly used
effective biocide (e.g., Torres et al., 2005), and compared the results with those of untreated
(control) wet extractions. We prepared 12 additional ice-wedge samples using the random cube
method for these tests (see Materials and Methods section). We found no significant differences
between the control and $HgCl_2$-treated wet extraction results for both $CH_4$ and $N_2O$ mixing
ratios (Figures 1e and 1f), indicating that the bias due to microbial activity during
approximately an hour of the melting-refreezing procedure is not significant. This is further
supported by tests on an additional 12 ice-wedge samples (using the random cube protocol)
treated with 2-bromo-ethane-sulfonate (BES), a specific methanogenesis inhibitor (e.g., Nollet
et al., 1997) (Figure A3). Similar to the $HgCl_2$-treated experiments, 25 μL of a saturated BES
solution was added to each sample flask. These additional tests were carried out only for $CH_4$.
The two-sided t-test for the $CH_4$ data indicates an insignificant difference between the two
results ($p > 0.9$). Data from individual sampling sites also do not show significant differences
($p > 0.9$ for the Alaskan samples and $p > 0.5$ for the central Yakutian samples).
According to microbial sequencing studies that have shown the presence of viable
microbes in permafrost and ground ice (e.g., Katayama et al., 2007), it is likely that culturable
microbes exist in the ice-wedge samples used in this study. However, considering that at least
14 days and up to 3 months of culturing was required to identify microbe colonies extracted
from ground ice (Katayama et al., 2007; Lacelle et al., 2011), our melt-refreeze time of an hour
was insufficient for microbial activity to resume and produce $CH_4$ and $N_2O$.

### 3.3. Dry extraction efficiency and gas mixing ratios


One limitation of our needle crushing dry-extraction technique is the inability to



completely extract gas from ice samples, because small ice particles and/or flakes placed in the
space between the needles are not fully crushed. The gas extraction efficiency of the SNU
needle crusher system has been reported as ~80–90% for polar ice core ice samples (Shin,
2014). However, the gas extraction efficiency has not been tested for ice-wedge samples.
Depending on the extraction efficiency, the needle crushing method could underestimate the
gas contents if the gas is not completely extracted. Another possible bias in the gas mixing
ratios arises if the $CH_4$ and $N_2O$ compositions are different between the crushed and uncrushed
portions of the ice-wedge samples.

230        To estimate the biases arising from incomplete gas extraction, we designed a series of

tests to identify the differences of the $CH_4$ and $N_2O$ mixing ratios and contents between the
crushed and uncrushed sample portions. Each ice-wedge sample that was randomly collected
was first crushed by the regular dry extraction procedure (by hitting it five times with the needle
system, 'hit5'), and the gas liberated from the sample was trapped in a sample tube. Then we
performed an additional 100 hits on the leftover ice ('hit100'), monitored the amount of
additional gas liberated, and trapped the additional gas in a separate sample tube. Comparisons
between the hit5 and hit100 results are summarized in Table 1.

238        Here we regard the ratio of gas content of hit100 to that of hit5 (hit100/hit5 ratio

hereafter) as a measure of the gas extraction efficiency of the needle crusher system. The results
demonstrate an average hit100/hit5 ratio of gas content of $0.40 \pm 0.07$ for the Zyryanka samples,
$0.24 \pm 0.07$ for the Bluff samples, and $0.14 \pm 0.11$ for the Cyuie samples (Table 1). Despite the
fact that the number of samples was limited, the ice-wedge samples from the different sites
show distinct hit100/hit5 ratios of the amount of extracted gas. However, we observed that the
leftover ice from the Bluff and Zyryanka samples were not well-crushed, even after 100 hits
with the needle crusher. This was especially true if the ice sub-samples contained soil
aggregates: the frozen soil aggregates were barely crushed. In contrast, the Cyuie samples were



247 relatively well-crushed, and the leftover samples were apparently finer-sized ice flakes. We

248 also observed that the hit100/hit5 ratios of gas content are highly variable within samples from

249 a particular site, implying that the extraction efficiency of the needle crusher not only depends

250 on site characteristics, but also on the individual ice sample hardness. When compared with the

251 dry soil content measured from the sub-samples used for wet extraction, no relationship was

252 observed between the dry soil content and the extraction efficiency. In addition, in the case of

253 samples uncrushed by the hit100 test, it is difficult to estimate the extraction efficiency using

254 the hit100/hit5 ratio of gas content, as the hit100 tests liberated only a marginal portion of gas

255 from these samples. This is because the large-sized uncrushed soil aggregates or particles may

256 have prohibited the needle crusher from crushing the small-sized ice flakes or grains. The

257 needles move up and down together, as they are fixed to a pneumatic linear motion feedthrough

258 device, thus if there is a sizable soil clod that cannot be crushed, it blocks the needle crusher

259 from moving further down. Therefore, we do not recommend using a needle crusher system to

260 measure gas contents in ice-wedge samples.

261   The hardness of the ice samples may also affect the gas mixing ratio analysis in the hit5

262 and hit100 procedures. The hit100/hit5 ratio of $CH_4$ mixing ratio of Bluff and Zyryanka

263 samples are less than 1 in four out of six samples, yielding an average of $0.9 \pm 0.5$. However,

264 all five samples from the Cyuie ice-wedges have ratios greater than 1, with an average of 4.7

265 $\pm 2.6$ (Table 1). The higher hit100/hit5 ratio of $CH_4$ mixing ratios of Cyuie samples indicates

266 that the gases extracted via the hit100 procedure have higher $CH_4$ mixing ratios than the gases

267 extracted via the hit5 procedure. Considering these results with those discussed previously, we

268 speculate that there are three ways gas can be trapped in ice-wedge ice: enclosed in bubbles,

269 adsorbed on soil particles, and entrapped in soil aggregates. The better-crushed leftover ice

270 flakes in the Cyuie samples may have allowed most of the gas in bubbles and part of the $CH_4$

271 molecules adsorbed on soil particles and/or trapped in microsites within soil aggregates to be





liberated. Thus, the hit5 $CH_4$ mixing ratios of the Cyuie samples may more reflect the gas
mixing ratios in bubbles, while the hit100 results reflect more of the contribution from gas
adsorbed on soil and trapped within soil aggregates than the hit5 results because soil-rich ice
has greater hardness than the soil-poor ice. If this is the case for the Cyuie samples, we can
infer that $CH_4$ is more concentrated in soil particles and in microsites within soil aggregates,
compared to in bubbles in the ice. This is partly supported by evidence that ice-wedge layers
exhibit relatively trace amounts of $CH_4$ compared to the surrounding permafrost soil layers
(Rivkina et al., 2007); however, this needs to be further evaluated by detailed microbial and
chemical analyses. In the meanwhile, in the Bluff and Zyryanka samples, the hit5 results reflect
the mixing ratios of the gases from the crushed portions, regardless of their origin: bubbles,
particle adsorption, or microsites in aggregates. Given that some of the Bluff and Zyryanka ice-
wedge samples were not fully crushed by the hit100 tests, it may require additional hits or
another extraction technique. Unlike $CH_4$, the $N_2O$ mixing ratios from the hit100 extractions
are higher than the hit5 in ten out of eleven samples, regardless of the sampling site. The
hit100/hit5 ratios of $N_2O$ mixing ratios of the Bluff and Zyryanka samples ($1.9 \pm 0.8$ on average)
are not significantly different ($p = 0.32$) from those of the Cyuie samples ($2.9 \pm 1.8$ on average).
This can probably be explained by the fact that the $N_2O$ mixing ratio is not necessarily higher
in soil-rich ice because $N_2O$ is an intermediate product in relatively oxic conditions, while $CH_4$
is produced strictly in anoxic conditions.

291         One may expect that a different crushing technique might be more suitable for ice-

wedge samples. However, none of the existing dry extraction techniques - centrifugal ice
microtome (Bereiter et al., 2013), mechanical grater (Etheridge et al., 1988), or ball-mill
crusher (Schaefer et al., 2011) is more advantageous for ice-wedge analysis compared to the
needle crusher system used in this study. The hard portion of ice wedges (e.g., frozen soil
aggregates, large soil particles) could easily damage the metal blades of the centrifugal ice



microtome and mechanical grater devices, or block the space within the ball-mill chamber,
limiting the movement of the milling balls.
It is worth noting that friction between stainless steels could produce $CH_4$ with carbon
from the damaged stainless-steel surface and hydrogen gas (Higaki et al., 2006). If needle
crushing causes contamination in this way, the dry extraction results should be affected by the
number of hits. To check the impact of the needle crushing procedure on ice-wedge $CH_4$ and
$N_2O$ measurements, we carried out blank tests by changing the numbers of hits from 5 to 100.
The results of these tests show no systematic offset among the experiments with different
numbers of hits (Figure A2), which implies that the crushing procedure does not affect the dry
extraction results for $CH_4$ and $N_2O$. Even though a small of contamination does exist, its effects
have already been subtracted via blank correction and taken into account in the overall error
estimation (see Appendix). Therefore, we consider that our findings are not artefacts of metal
friction during crushing.
To summarize, from the hit5 and hit100 comparison tests, we found that 1) the needle
crusher method is not able to fully crush the ice-wedge ice samples and thus is unsuitable for
measuring gas contents in a unit mass of ice, and that 2) weak crushing (e.g., a small number
of hits by the needle crusher system) may better reflect gas mixing ratios of the soft parts of
the samples (such as air bubbles) than strong crushing (e.g., a greater number of hits).





**Table 1.** Results of dry extraction tests with 5- and additional 100 times hitting ice-wedge samples, denoted as 'hit5' and 'hit100', respectively. 'hit100/hit5' is the ratio in extracted gas content or gas mixing ratio of 'hit100' to 'hit5' cases. Also shown are gas content results from both experiments, where the hit100 values are given both in the unit of ml kg$^{-1}$ at STP conditionsand μmol/kg (in parenthesis). It should be noted that the 'hit100' gas content results indicate the additional amount of gas extracted after 'hit5' crushing and evacuation.

| Site Location | Sample | gas content | | | | CH$_4$ mixing ratio | | | | N$_2$O mixing ratio | | | |
|---|---|---|---|---|---|---|---|---|---|---|---|---|---|
| | | Wet control | Dry hit5 | Dry hit100 | hit100/hit5 | Wet control | Dry hit5 | Dry hit100 | hit100/hit5 | Wet control | Dry hit5 | Dry hit100 | hit100/hit5 |
| | | ml/kg | ml/kg | ml/kg | | ppm | ppm | ppm | | ppm | ppm | ppm | |
| Zyryanka, Northeastern Siberia | Zy-A-W1-D | 20.2 | 13.1 | 6.3 | 0.48 | 6138 | 3713 | 2721 | 0.7329 | 11.37 | 9.10 | 10.15 | 1.12 |
| | Zy-F-1 | 13.5 | 8.1 | 3.4 | 0.42 | 1080 | 655.6 | 173.5 | 0.2646 | 1.57 | 2.81 | 2.65 | 0.942 |
| | Zy-A-W1-Low | 30.6 | 27.8 | 8.0 | 0.29 | 4309 | 5073 | 4818 | 0.9497 | 2.07 | 0.69 | 2.02 | 2.9 |
| | Zy-B-Low-B | 29.1 | 23.9 | 10.0 | 0.418 | 18030 | 21010 | 35290 | 1.680 | 5.37 | 5.32 | 15.36 | 2.89 |
| Northern Alaska | Bluff03-IW1 | 13.2 | 12.2 | 2.6 | 0.21 | 44160 | 25230 | 12240 | 0.4851 | 5.58 | 2.36 | 4.93 | 2.09 |
| | Bluff06-B3 | 20.1 | 20.9 | 5.6 | 0.27 | 558.7 | 164.2 | 219.5 | 1.337 | 3.74 | 18.78 | 30.14 | 1.605 |
| Cyuie, Central Yakutia | CYC-01-B | 18.0 | 21.7 | 7.1 | 0.33 | 18.0 | 18.3 | 25.4 | 1.39 | 1.55 | 1.60 | 2.59 | 1.62 |
| | CYB-04-C | 20.9 | 30.7 | 1.5 | 0.049 | 20.2 | 48.4 | 165.6 | 3.42 | 0.71 | 0.65 | 2.96 | 4.5 |
| | CYB-03-A | 19.7 | 23.7 | 1.0 | 0.041 | 20.5 | 21.5 | 67.1 | 3.12 | 0.91 | 1.01 | 1.06 | 1.05 |
| | CYB-02-A | 32.0 | 25.5 | 1.9 | 0.073 | 29.1 | 18.7 | 159.8 | 8.55 | 1.00 | 0.58 | 3.19 | 5.5 |
| | CYC-03-B | 22.6 | 15.7 | 3.3 | 0.21 | 20.3 | 13.9 | 94.5 | 6.80 | 1.40 | 0.65 | 1.08 | 1.7 |




### 3.4. Residual gas mixing ratios and contents after wet extraction


To examine how well the gas is extracted by wet extraction, we applied the dry
extraction method to refrozen ice-wedge samples after wet extraction. We first prepared
degassed ice-wedge samples that had undergone repetitive wet extractions (wet-degassed ice
hereafter). Once the wet extraction experiments were completed, we repeated two cycles of
melting-refreezing and evacuation procedures to degas the ice melt. After degassing by a total
of three cycles of wet extraction and evacuation, the outermost surfaces (~2 mm) of the wet-
degassed ice were trimmed away in the walk-in freezer at SNU on the morning of experiments.
The wet-degassed ice was then inserted into the needle crusher and the crusher chamber was
evacuated. A specific amount of standard air was injected. Then, the wet-degassed ice samples
were hit 20 or 60 times by the needle crusher. The amount of gas and gas mixing ratio of the
additionally extracted gas from the wet-degassed ice are shown in Figure 2 and Table A1.
The tests using the wet-degassed ice show an additional gas extraction of ~12 to 20 ml
$kg_{ice}^{-1}$, which is ~43 to 88% of the amount of gas extracted during the initial wet extraction.
The additionally extracted gas from the dry extraction is referred to as residual gas hereafter.
This is remarkably in contrast to the less than 1% residual fraction of the SNU wet extraction
system for ice from polar ice sheets. If such a considerable amount of gas is left intact by
repeated wet extractions, the composition of the additional gas is important to understand how
much the conventional wet extraction results are biased.
Figure 2 and Table A1 show the mixing ratios and contents of $CH_4$ and $N_2O$ in the
residual gas. The mixing ratios of the residual gas were estimated using mass balance
calculations with observed mixing ratios and the amounts of the injected standard and extracted
residual gas. The $CH_4$ mixing ratios of the residual gas range from 10.37 to 23.78 ppm, which
is similar to the range of the wet extracted gas. This evidence indicates that $CH_4$ in ice-wedges
cannot be fully extracted by a melting-refreezing procedure. We suspect two possible reasons





for this: (1) During wet extraction, the ice-wedge samples melted and the soil particles settled
at the bottom of the sample flask without any physical impact to the soil particles, causing the
adsorbed $CH_4$ molecules on the soil particles to remain adsorbed. (2) During refreezing, the
soils accumulated at the bottom of the flasks are crumpled around the centre of the refrozen
ice, because the sample flasks are chilled from outside, which facilitated gas entrapment within
the frozen soil aggregate. In contrast, the $N_2O$ mixing ratios of the residual gas exhibit very
low values compared to those from the initial wet extraction (Figure 2 and Table A1). These
results imply that most of the $N_2O$ in ice wedges is extracted by three melting-refreezing cycles,
such that only a small amount of $N_2O$ is left adsorbed or entrapped in ice-wedge soils.
In this section, we found that a certain amount of gas remained in ice wedges, even after
three cycles of wet extraction, which is extractable instead by needle crushing. This implies
that, unlike polar ice cores, wet extraction of ice-wedges does not guarantee near-complete gas
extraction, and therefore, precise measurements of the gas content of ice wedges are difficult
to obtain. The difficulty in measuring gas content imposes a large uncertainty in estimating
$CH_4$ and $N_2O$ contents. Furthermore, we found that the residual gas has a similar order $CH_4$
mixing ratio as the gas extracted by initial melting-refreezing, indicating that a comparable
amount of $CH_4$ still remains unextracted in ice-wedges. Hence, a novel extraction method is
required to produce reliable gas content and gas mixing ratios in ice wedges. In contrast, our
results show that the $N_2O$ content of the residual gas is at trace levels, which may suggest that
most of the $N_2O$ in ice-wedges is extractable during initial melting-refreezing. Therefore, wet
extraction could be applicable for estimating the $N_2O$ content of ice wedges. However, given
that the above evidence resulted from three consecutive cycles of melting-refreezing and
evacuation, it is unclear how many melting-refreezing cycles are required to extract most of
the $N_2O$ from ice wedges. Our findings imply that previous estimates of $CH_4$ budget in ground
ice based on wet extraction principle (e.g., Boereboom et al., 2013; Cherbunina et al., 2018)





might have been underestimated, and that the $CH_4$ production within subfreezing permafrost
environment could be larger than previously estimated. Future study should be devoted to a
novel extraction method which is able to extract gas molecules from ice effectively.

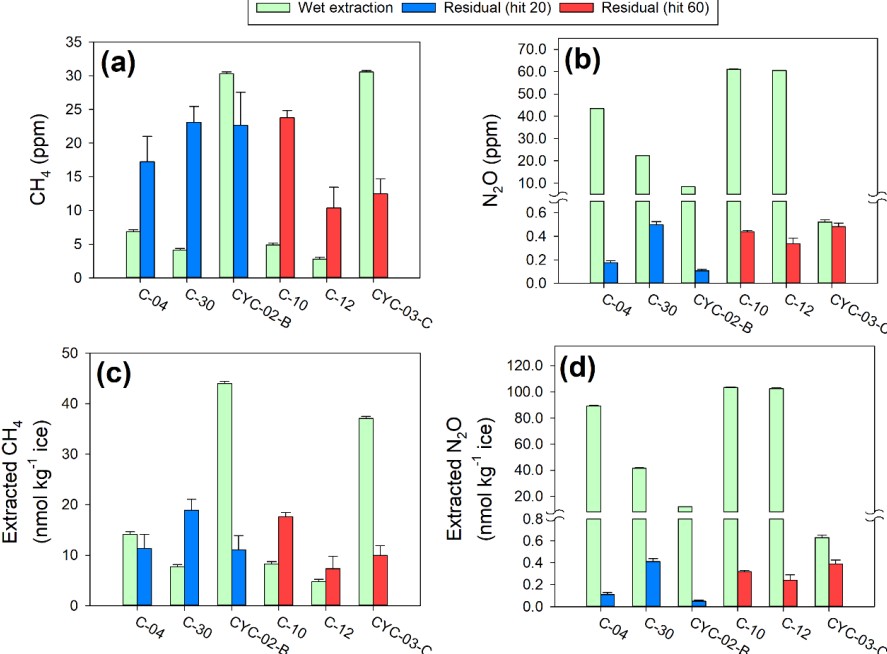

**Figure 2. Comparison of wet-extracted gas and residual gas for $CH_4$ and $N_2O$ mixing ratios (a and b) and contents (c and d).** The residual gas was extracted from the dry extraction method using the wet-degassed ice samples. The light green bars show the results of initial wet extraction, and the blue and red bars indicate the dry extraction of wet-degassed ice with 20- and 60-times hitting, respectively.


**4. Conclusions**
In this study we carried out comparisons between wet and dry extractions, between
untreated and biocide-treated wet extractions, and gas extraction from the easily to extract and
difficult to extract parts of ice-wedge ice to better understand the characteristics of each
extraction method, in order to adequately analyse $CH_4$ and $N_2O$ mixing ratios and gas contents
from permafrost ice wedges. Based on these comparisons, our major findings are summarized
as follows:

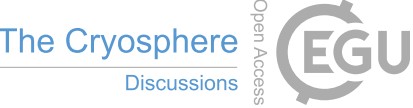

1) Existing wet and dry extraction methods allow gas extraction from the soft parts of ice (e.g., ice bubbles) and show insignificant differences in $CH_4$ and $N_2O$ mixing ratios.

2) Wet extraction results are unlikely to be affected by microbial production of $CH_4$ and $N_2O$ during the melting-refreeze procedure.

3) Both dry and wet extraction methods are not able to fully extract gas from ice wedge samples, presumably due to gas adsorbed on soil particles or enclosed within soil aggregates, which may have different gas mixing ratios compared to the gas in bubbles. Further research is required to develop a proper method to quantify and extract adsorbed and enclosed gases. In the meantime, we propose that both existing techniques may be suitable for gas mixing ratio measurements for bubbles in relatively soft ice wedges. Exceptionally, the $N_2O$ content in ice wedges may be measured by using repeated wet extractions, but this is not the case for determining the $N_2O$ mixing ratio.

4) Our results indicate that previous estimates of ground ice $CH_4$ and $N_2O$ budget might be underestimated, implying that the greenhouse gas production in subfreezing environment of permafrost is larger than our current understanding.

5) Our finding indicates that the saturated NaCl solution is unnecessary to prevent microbial activity during melting, as employed by, e.g., Cherbunina et al. (2018). However, it remains as an open question how effectively the adsorbed gas molecules can be extracted by the method.



**Appendix. Systematic blank correction and uncertainty estimation**

Since the SNU dry extraction systems, including the sample tubes, were originally designed for $CO_2$ measurements from polar ice cores, these systems have not been tested for $CH_4$ and $N_2O$ analysis. We therefore carried out a series of tests to estimate the systematic blank, which is defined here as blanks.

The systematic blanks were tested with bubble-free ice (BFI) and standard air in a cylinder calibrated by NOAA. The BFIs were prepared as described in Yang et al. (2017). A major difference is that the BFI block was cut into small BFI pieces of 3–4 g, to mimic the random cube sampling protocol (see Materials and Methods section in the main text). The systematic blanks for the dry extraction method were tested as follows. A total of ~45 g of BFI cubes was placed into the crushing chamber, sealed with a copper gasket, and evacuated until the gas pressure inside the chamber dropped lower than ~60 mTorr, because of the vapor pressure formed by sublimation of the BFI. After evacuation was completed, standard gas was injected into the crushing chamber. The amount of standard injected was controlled by a volume calibrated vacuum line in the dry extraction system. Then the BFI samples were hit with the needle system 5 to 100 times, and the gases in the chamber were passed through a water trap and cryogenically pumped into the sample tubes, using the He-CCR. The number of hits did not significantly affect the systematic blank (Figure A2) and the regression curve for blank correction was fitted to the entire set of data points (red dashed curve in Figure A1).

For the wet extraction, a total of ~45 g of BFI cubes was placed into each sample flask. The flasks were connected to the wet extraction line and sealed with a copper gasket, then evacuated. Once a vacuum was established, a known amount of standard gas was injected into each flask and the flasks were submerged into a warm water bath for ~40 min to melt completely. The flasks were then submerged into the cold ethanol bath, which was chilled to -80°C, to refreeze. For the $HgCl_2$ and Sodium 2-bromo-ethane-sulfonate (BES) treated

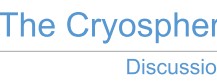
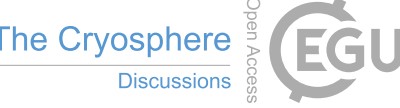

experiments, we first prepared the saturated solutions of $HgCl_2$ and BES at room temperature
(20°C) and added 24 μL of $HgCl_2$ or 20 μL of BES solution into the empty flasks in a fume
hood. Then we placed the flasks in a deep freezer, maintained at -45°C for 20 min, to freeze
the solutions before the BFI pieces were placed.

432        The results of the blank experiments are shown in Figure A1. The systematic blanks

appear to be inversely correlated with the gas pressure in the sample tube. The systematic blank
test results were fitted using exponential regression curves (dashed lines in Figure A1), and
these regression curves were then used for systematic blank correction in our ice-wedge sample
analyses.

437        To calculate uncertainties of the blank corrections, the blank test data were fitted with

exponential regression curves (Figure A1). The root-mean-square-deviations (RMSD) of the
data from the regression curves are taken as the uncertainties of blank corrections (Figure 1).
Since the ice-wedge data used in this study showed the pressure in GC sample loop of about 8
~ 50 torr, the RMSD were estimated from the blank test data within this pressure range. The
uncertainty of the gas content measurement is calculated by error propagation from those of
pressure, line volume, and mass of ice samples.

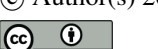

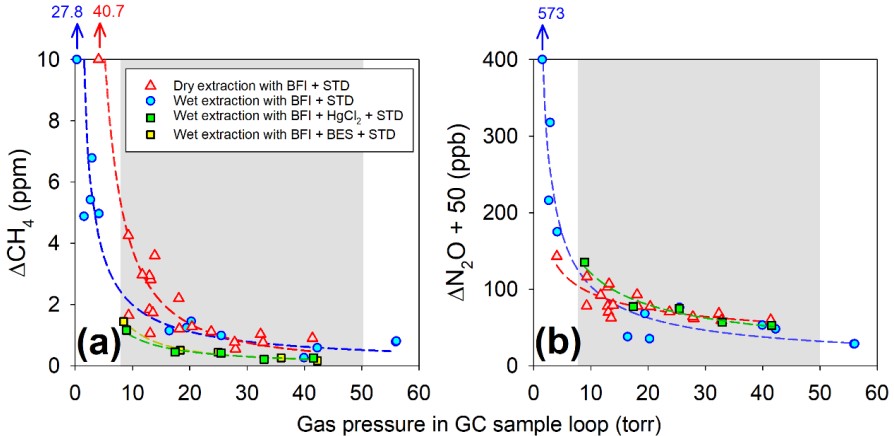


**Figure A1.** Systematic blank of the needle crushing (dry extraction) and melting-refreezing (wet extraction) methods for (a) $CH_4$ and (b) $N_2O$ measurements in control and biocide ($HgCl_2$) treated experiments. Also plotted are the $CH_4$ blanks of BES-treated wet extractions. The dashed lines represent exponential regression curve fittings. Note that all data are plotted against the amount of gas trapped in the sample tube, presented here as the pressure in the GC sample loop when the sample gas is expanded. The grey shaded areas indicate the range of ice-wedge samples used in this study (see main text). The big-delta ($\Delta$) notion in the y-axes indicate the offset from the values of the standard used.

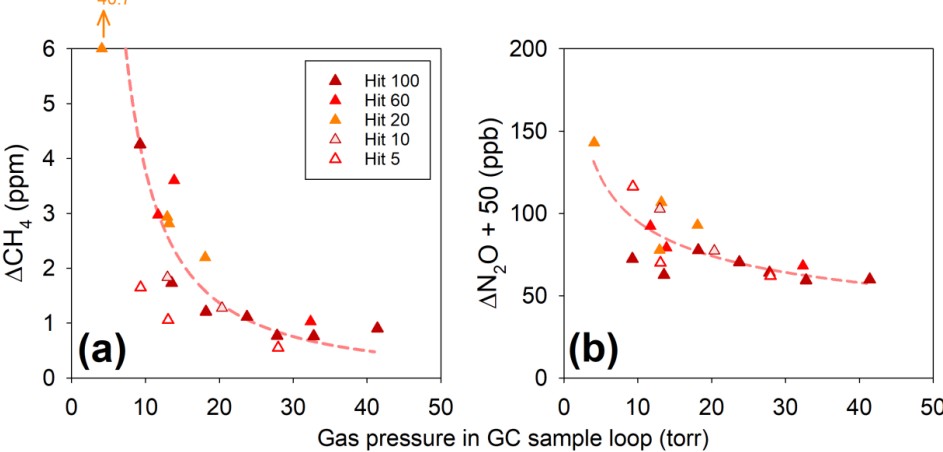

**Figure A2.** Influence of different number of hitting on the systematic blank of the needle crushing (dry extraction) system for (a) $CH_4$ and (b) $N_2O$ measurements. Note that all data are plotted against the amount of gas trapped in the sample tube, presented here as the pressure in the GC sample loop when the sample gas is expanded (see main text). The big-delta ($\Delta$) notion in the y-axes indicate the offset from the values of the standard used.








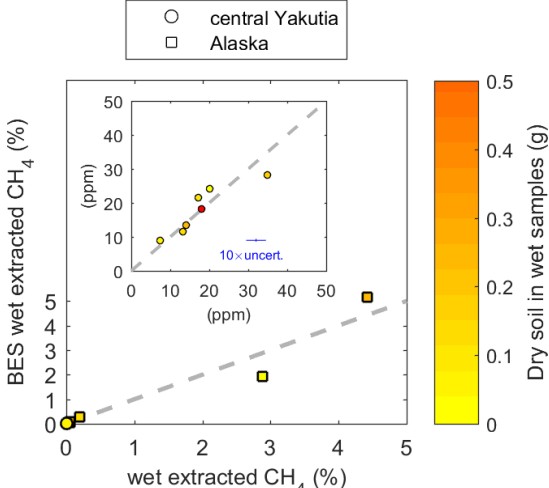

**Figure A3.** Comparison between control- and BES-treated wet extraction results for $CH_4$. The sampling area is indicated by different symbols. The color of each data point indicates the dry soil weight in the subsamples used in control wet extraction. The grey dashed lines are 1:1 reference line.







**Table A1.** Comparison of results from extracted gas from the conventional wet extraction method and the residual gas in ice after 3-times wet extraction. The residual gas was extracted by a needle crusher (see section 3.4 for details of the mehtods)

| Site location | Sample | soil content | Wet extraction | | | | | Residual gas | | | | |
|---|---|---|---|---|---|---|---|---|---|---|---|---|
| | | | gas content | CH₄ mixing ratio | N₂O mixing ratio | CH₄ content | N₂O content | gas content | CH₄ mixing ratio | N₂O mixing ratio | CH₄ content | N₂O content |
| | | wt. % | ml/kg | ppm | ppm | nmol/kg | nmol/kg | ml/kg | ppm | ppm | nmol/kg | nmol/kg |
| Churapcha, central Yakutia | C-10 | 0.524 | 37.9 | 4.9 | 61.13 | 8.3 | 103 | 16.6 | 23.8 | 0.437 | 17.6 | 0.324 |
| Churapcha, central Yakutia | C-30 | 1.03 | 41.7 | 4.1 | 22.28 | 7.7 | 41.5 | 18.4 | 23 | 0.50 | 19 | 0.41 |
| Cyuie, central Yakutia | CYC-03-C | 1.09 | 27.2 | 30.5 | 0.52 | 37.1 | 0.63 | 17.9 | 12.5 | 0.48 | 10.0 | 0.39 |
| Churapcha, central Yakutia | C-04 | 1.38 | 46.0 | 6.9 | 43.46 | 14 | 89.2 | 14.7 | 17 | 0.17 | 11 | 0.11 |
| Cyuie, central Yakutia | CYC-02-B | 1.12 | 32.5 | 30.3 | 8.34 | 44.0 | 12.1 | 11.0 | 23 | 0.11 | 11 | 0.053 |
| Churapcha, central Yakutia | C-12 | 0.370 | 38.0 | 2.8 | 60.47 | 4.8 | 103 | 15.9 | 10 | 0.34 | 7.3 | 0.24 |





**Data availability**

The data will be uploaded on the public data repository of Pangaea after publication.

**Author contributions**

JWY and JA conceived the research and designed the experiments. GI, JA, KK, and AF drilled the

ice-wedge ice samples from Alaska and Siberia. JWY, JA, SH, and KK conducted the laboratory

experiments. JWY and JA led the manuscript preparation with inputs from all other co-authors.

**Competing interests**

The authors declare no conflict interest.

**Acknowledgements**

The authors greatly acknowledge those who contributed to collect ice-wedge ice samples. We

thank Gwangjin Lim and Jaeyoung Park for their help in sample preparations and gas extraction

experiments, and Min Sub Sim for kind advice on inhibition experiment for methanogen.

**Financial support**

This project was supported by the Basic Science Research Program through the National Research

Foundation of Korea (NRF) (NRF-2018R1A2B3003256) and the NASA ABoVE (Arctic Boreal

and Vulnerability Experiment; grant no. NNX17AC57A).





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
