# Peer review of "Ji-Woong Yang1\*, Jinho Ahn1, Go Iwahana2, Sangyoung Han1, Kyungmin Kim1\*\* and Alexander Fedorov3,4"

_The Cryosphere, 2019_

## Referee Comment (RC1) · Anonymous Referee #1 · 4 Nov 2019

**1) GENERAL COMMENTS**

This methodological, brief communication paper reports on a comparison of different techniques (wet *vs*. dry extraction, with or without biocide to test microbial contamination) to extract gas ($CH_4$ and $N_2O$) from ice wedges of Alaska and Siberia. The authors report that tested methods yield good results for the easily extractable gas fraction (bubbles), but this is not so convincing for the adsorbed phase or gas contained within soil aggregates. One of the main conclusions, therefore, is that current estimates of ground-ice gas budgets are likely underestimated, as a fraction of produced gases are not taken into account. For me, this is the main take-home message.

It appears as an interesting short paper, although the methodology used is not in my immediate field of expertise. To my knowledge, this manuscript does not have major flaws that should ultimately prohibit its publication. It is generally well written and easy to read. I have however a few points to mention that preclude acceptance for publication as is:

1) I am not convinced, for now, of the general, broad-audience impact of the manuscript. Does it really « *report new developments, significant advances, and novel aspects of experimental and theoretical methods and techniques which are relevant for scientific investigations within the journal scope* »? (https://www.the-cryosphere.net/about/manuscript_types.html). The authors have not convinced me that this work is new, innovative or represent a major advancement that is relevant to the community at large. They rather suggest that a future, novel extraction method might provide better results. I am also looking forward to that. This work might be useful for a small specialized group, however. Furthermore, the conclusion about underestimation of current gas budgets in ice-wedge terrains is itself interesting and timely.
2) I understand that this is a brief communication and that the number of figures/tables is limited. However, it is really unfortunate that there is no map of the many study sites, and no picture or illustration of field sampling procedures, as well as lab instruments (especially the 'needle crushing system'). It would greatly help to have visual support for such investigations.
3) Several sentences contained in the results/discussion section are in fact related to methods. I mention examples in the specific comments section below. The structure of the main text should therefore be re-aligned, so that methods sentences are in the methods section.
4) Finally, some statements and conclusions in the main text are either not accompanied by a mention to the results or figure(s) they come from, or not supported by literature reference(s).

Overall, I cannot accept this manuscript for publication as is. If the authors are willing to make major revisions (general points above and specific comments below), I would be happy to review a revised version of the manuscript.

**2) SPECIFIC COMMENTS AND EDITORIAL SUGGESTIONS**

P= page number, L = line number.

P1, L29. To avoid repetition (soil): choose either « Permafrost preserves large amounts of soil carbon and nitrogen… », or « Permafrost soils preserve large amounts of carbon and nitrogen… ».

P2, L30. I suggest adding 'temporarily': «… temporarily removing this frozen carbon… ».

P2, L30-31. (C) and (N) should be put at the beginning of the section (P1, L29), i.e. the first time that the words 'carbon' and 'nitrogen' are mentioned.

P2, L35. « … which in turn can trigger positive feedbacks… ».

P2, L38-40. This might be true for Yedoma regions (eastern Siberia, Alaska, Yukon), but not all permafrost it necessarily ice-rich. It should be specified in the paragraph, otherwise we have the impression that permafrost all over the Arctic contains 40-90% of ground ice.

P2, L40. I suggest adding 'Pleistocene': « … volume of Pleistocene ice-rich permafrost, or Yedoma ».

P2, L42. « … evidence for in-situ microbial aerobic respiration… ». Why just 'aerobic' conditions? This might be relevant for CO2 production, but CH4 and N2O are generally produced under 'anaerobic' conditions, or both oxic-anoxic.

P2, L43-44. « … detailed information on in-situ biogeochemical processes responsible for GHG production… ». Which biogeochemical processes? Methanogenesis? Respiration? Other processes?

P3, L58-59. «… because ice wedges are one of the most abundant morphological features… ».

P3, L73-79. For the reader not familiar with the study sites and ground-ice sampling protocols in permafrost landscapes, I strongly suggest adding 1) a map of the study sites (Siberia and Alaska); 2) pictures of an outcrop and sample collection (drilling). This way, the reader would have a much better idea of what the samples and sites look like.

P4, L80-97 and L88-95. Again, all these descriptions and distances would make much more sense if they were accompanied by a map (with sampling sites labeled on the map).

P4, L87. «… on the first terrace of the river… ». Do the authors mean the younger (i.e. lower) terrace?

P4, L96. « The ice-wedge ice… ». This phrase is weird. Suggestions: « The ice from ice wedges is different from polar ice cores, in that… », or « Wedge ice is different…».

P5, L107. «… 8~13 g of ice sample were crushed… ».

P5, L121-123. Which year for this modern air sample? Please specify.

P7, L175. I suggest 'thoroughly' or 'vigorously' instead of 'well'. («… shaken flasks were shaken thoroughly/vigorously… »

P8, L187-189. What is meant by this statement about the heterogeneous distribution of samples? How is it shown on Fig. 1?

P9, L195-196. This is a busy figure, see comments below (section 3) FIGURES). Some elements could be removed to enhance clarity.

P10, L208-211. This reads more like methods, not a results and discussion section.

P10, L211-212. How about the Eastern Siberia samples? (triangles in Fig1) Did they also « not show significant differences » between the two sets of tests?

P11, L224. «… polar ice core samples… » (remove the 2nd 'ice')

P11, L230-237. This reads more like a paragraph about methods.

P11, L236-237. This information would be better displayed and more appealing in a figure.

P12, L250-252. This statement is based on what result? Can we see this displayed somewhere in a figure/table? If yes, please refer to it in the text.

P12, L255-260. For the reader not familiar with the needle crushing system, a picture or a sketch of what the apparatus looks like might help. The sentences in this paragraph would be more easily understood.

P13, L280-282. Please refer to results (figure or table) to support this statement.

P13, L288-290. Please support this statement by relevant references. In fact, the simple association N2O=oxic / CH4=anoxic is not entirely and always true. For example, N2O production has been recorded under both oxic and anoxic conditions (Gil et al. 2017; *Global Biogeochemical Cycles*), as well as CH4 production from oxic waters (Grossart et al. 2011; *PNAS*). It depends on several parameters, including local hydrology (e.g., water-logged soils). This should be acknowledged in the text.

P13, L292 to P14, L298. This statement is highly speculative. Unless I missed something, this was not tested for real in this study. This paragraph should be supported by real data or removed.

P14, L302-303. Again, this is methodology, not results/discussion.

P16, L320-330. Again: methodology, not results/discussion.

P17, L350-352. This is indeed interesting. Do we know why N2O appears to be more extractable (or less present in the residual adsorbed phase) than CH4, at least based on the wet extraction technique? Was this already observed elsewhere and reported in the literature?

P18, L375. «… easy to extract… »

P19, L397. « Our findings indicate that ... »

**3) FIGURES AND TABLES**

Figure 1, P9.
   a) This is a pretty busy figure. We don't necessarily need the 3 legends (identical) in the middle. By removing them, more space could be created to enlarge the graphs a bit, because for now they are quite small. Also: what is the purpose of the insets (a-b, e-f)?
   b) I don't get the thing about the error bars (in blue). Are these 5x, 100x or 500x larger or smaller in 'real life' than displayed on the graphs? Not clear.
   c) Explain what does 'hit5' mean.

Table 1, P15.
   a) It is not explained why the hit100/hit5 ratios for gas content (6[th] column) are much lower for most of the central Yakutia samples (Cyuie), compared to the other sites? This is indeed interesting, but why? Less soil aggregates in ice-wedge samples from this site, so relatively more bubbles and thus more extracted gas?

Figure 2, P18.
   a) Where do the samples come from? CYC-02-B and CYC-03-C likely refer to Central Yakutia (Cyuie), but what about the other samples (C-04, C-30, C-10, C-12)? Please specify somewhere.

---

## Referee Comment (RC2) · Anonymous Referee #2 · 26 Nov 2019

**1) GENERAL COMMENTS**

The paper reports novel aspects of experimental methods of gas extraction techniques. Currently, there are no works that conducted such a comparison, which has long been needed   due to huge number of gas measurements in permafrost area conducted recently.  Unfortunately, there is still no unified method for carrying out gas extraction that leads to the impossibility and impropriety of comparison the results obtained by different research groups.

The authors tested conventional wet and dry gas extraction methods for ice wedges coming to conclusion that current estimates of ground-ice gas budgets are likely underestimated. They found insignificant effects of microbial activity during wet extraction and significant difference in extraction results from polar ice cores and ice wedges.  Therefore the manuscript is of big interest for scientific community and contributes to changing our scientific understanding of a subject as it is has to be for TC.

The results are presented in well-structured way and the paper is easy to read.

However, there are a few general suggestions that could improve the article:
1)  I suggest adding a map of study sites, maybe some geological sections to get a better idea of the location and structure of ice wedges. Are they all Pleistocene?
2) Besides   it is really necessary to include the schemes of gas extraction procedures (both wet and dry techniques as well as the experiment on dry extraction efficiency and on residual gas contents after wet extraction.  Due to the limitation of the number of figures both 1 and 2 can be added as supplementary material.
3) Since the article is devoted to the comparison of methods, it would be useful to estimate the limits of applicability of the methods and measurement errors.
4) It is necessary to add initial data  on gas content and  $CH_4$ and $N_2O$  mixing ratios as supplementary material to prove you main result about  the same effectiveness of wet and dry extraction methods. As I see now from the Table 1 there can be  2  times difference  (up to 20000 ppm) for $CH_4$

So the article is of big interest but needs major revision to be accepted and I would be happy to review a revision of the paper.

**2) SPECIFIC COMMENTS AND EDITORIAL SUGGESTIONS**

P.2 L50-51 «…ice sample was melted in a saturated sodium chloride (NaCl) solution, in order to minimize microbial activity and gas dissolution (Cherbunina et al., 2018 and references therein)». I found no mention  in the article that NaCl was used in order to minimize microbial actvity , is it really there?

P.3 L72. Please specify the size of the samples, add the site map and geological sections with sampling location

P.4 L104. Please include the schemes of gas extraction procedures

P.5 L108  Why it is used precisely 5 times, not 100, can you reason it  somehow?

P.6 L144  It is not clear where did you get the dry soil mass before the extraction.  «Taking the dry soil mass of the analysed samples (0.33 g) into account, we added 24 μL  of saturated $HgCl_2$ solution (at 20°C) to the sample flasks» Were there used the data on dry soil mass from the other

samples? Because later you say « Dry soil content was measured using the leftover meltwater from the control-wet extraction tests. »

P.12 L249. Please specify what do you mean by «ice hardness» here. As in L 249 «the extraction efficiency of the needle crusher not only depends on site characteristics, but also on the individual ice sample hardness», and later L. 251 «no relationship was observed between the dry soil content and the extraction efficiency», but L.274 « soil-rich ice has greater hardness than the soil-poor ice». I guess this is the matter of «soil aggregates» as you mention later, so the hardness in this case is defined by this parameter? Is it posiible to quantify this?

P.12 L255. Please specify the size range for «This is because the large-sized uncrushed soil aggregates or particles may have prohibited the needle crusher from crushing the small-sized ice flakes or grains». As the presence of the aggregates is one of the main limits to use the technique, is it possible to make at least a rough estimate of the amount of gas that can remain there?

P.12 L259. «Therefore, we do not recommend using a needle crusher system to measure gas contents in ice-wedge samples». Can you estimate the efficiency of the method in % in the same way it has been done for polar ice core ice samples (80–90% )(Shin, 2014)? As I see from the Table 1 the procedure « Hit5+Hit100» in most cases allows to extract more gas then the wet method even if uncrushed aggregates still occur. Can you recommend using dry extraction method in this modification?

P.13 L272. Since you talk about gas in bubbles here : «the hit5 $CH_4$ mixing ratios of the Cyuie samples may more reflect the gas mixing ratios in bubbles, while the hit100 results reflect more of the contribution from gas 2 adsorbed on soil and trapped within soil aggregates than the hit5 results» and further, may be it would be useful to get the data on ice porosity to compare with the results of extracted volume of gas since the volume of gas normalized to layer pressure approximately corresponds to porosity.

P.16 L320. Please explain if I understand correctly the next paragraph:

«To examine how well the gas is extracted by wet extraction, we applied the dry extraction method to refrozen ice-wedge samples after wet extraction. We first prepared degassed ice-wedge samples that had undergone repetitive wet extractions (wet-degassed ice hereafter). Once the wet extraction experiments were completed, we repeated two cycles of melting-refreezing and evacuation procedures to degas the ice melt. After degassing by a total of three cycles of wet extraction and evacuation, the outermost surfaces (~2 mm) of the wet degassed ice were trimmed away in the walk-in freezer at SNU on the morning of experiments. The wet-degassed ice was then inserted into the needle crusher and the crusher chamber was evacuated. A specific amount of standard air was injected. Then, the wet-degassed ice samples were hit 20 or 60 times by the needle crusher.»

After the first freezing-melting cycle the sample gas in the headspace of the flask was collected and gas content, $CH_4$ and $N_2O$ ratio was measured. Then the two cycles were conducted. ( and my question here is what happened to the gas in the flask-was it collected and measured or just evacuated) and next step was measuring the gas content in degassed ice through needle-crusher procedure.

P.16 L328. Explain please why were such parameters chosen if in the previous dry extraction procedure you used 5 and 100 hits: « Then, the wet-degassed ice samples were hit 20 or 60 times by the needle crusher»

P.16 L331. The tests using the wet-degassed ice show an additional gas extraction  of 43 to 88% of the amount of gas extracted during the initial wet extraction. I suggest to add this information to the conclusions as it is of big significance as well as if I get it right the best way of degassing the sample according to your manuscript is to combine three cycles of wet extraction with dry extraction for the residual  gas. I think this has to be one of the main conclusion.

P.19 L391. Please specify what do you mean by «relatively soft ice wedges».

P.19 L 392   It seems to me that you have very good results of applying the method of three-times wet extractions+residual gas extracted by a needle crusher for $N_2O$ and I don't get why there is in conclusions «Exceptionally, the N2O content in ice wedges may be measured by using repeated wet extractions, but this is not the case for determining  the N2O mixing ratio»

**3) FIGURES AND TABLES**

**Table 1. Add a column of dry soil content as in table 2**

**The table with the data used for Fig.1 need to be added to get the difference between wet and dry method results.**

---

## Author Comment (AC1) · 14 Jan 2020

**We would like to thank the two anonymous reviewers and the Editor for their careful reviews of our manuscript. Below we present our point-by-point responses to all of the comments and annotations. The original comments from the reviewers are shown in black, while our responses are presented in red. Where indicated, additions to the manuscript are shown in blue and removed passages shown in red strikethrough font.**

================================================================================

**Comments from the reviewer #1 and our responses:**

**1) GENERAL COMMENTS**

This methodological, brief communication paper reports on a comparison of different techniques (wet *vs*. dry extraction, with or without biocide to test microbial contamination) to extract gas ($CH_4$ and $N_2O$) from ice wedges of Alaska and Siberia. The authors report that tested methods yield good results for the easily extractable gas fraction (bubbles), but this is not so convincing for the adsorbed phase or gas contained within soil aggregates. One of the main conclusions, therefore, is that current estimates of ground-ice gas budgets are likely underestimated, as a fraction of produced gases are not taken into account. For me, this is the main take-home message.

It appears as an interesting short paper, although the methodology used is not in my immediate field of expertise. To my knowledge, this manuscript does not have major flaws that should ultimately prohibit its publication. It is generally well written and easy to read. I have however a few points to mention that preclude acceptance for publication as is:

1) I am not convinced, for now, of the general, broad-audience impact of the manuscript. Does it really « *report new developments, significant advances, and novel aspects of experimental and theoretical methods and techniques which are relevant for scientific investigations within the journal scope* »? ([https://www.the-cryosphere.net/about/manuscript_types.html](https://www.the-cryosphere.net/about/manuscript_types.html)). The authors have not convinced me that this work is new, innovative or represent a major advancement that is relevant to the community at large. They rather suggest that a future, novel extraction method might provide better results. I am also looking forward to that. This work might be useful for a small specialized group, however. Furthermore, the conclusion about underestimation of current gas budgets in ice-wedge terrains is itself interesting and timely.

➔ We believe that our manuscript has broad implication to the large community working on permafrost-climate interaction. Permafrost thawing is a major potential threat of future global warming, expected to input large amount of greenhouse gas (GHG) to the atmosphere. Thus, quantification of the permafrost GHG budget is important for better projection of future climate change. There are a growing number of works reporting methane ($CH_4$) and nitrous oxide ($N_2O$) concentration either in permafrost or ground ice. Nonetheless, there has been no consensus on gas extraction methods, nor they have been tested properly. To our best knowledge, this manuscript is the first attempt to test different gas extraction methods to understand the applicability and limitations of both techniques. We believe that our study makes a significant contribution to the literature because our experiments show that existing methods allow gas extraction from the soft parts of ice, however gas adsorbed to or trapped in soil particles may not be extracted, affecting the measure GHG contents and their mixing ratios. In addition, we reveal that the microbial activities have insignificant impact on the wet extraction results. Although our manuscript does not deal with <*new development*>, we do believe that all these findings provide <*significant advances*> and <*novel aspects of experimental methods*>. Development of a new technique is far beyond the scope of this manuscript.

2) I understand that this is a brief communication and that the number of figures/tables is limited. However, it is really unfortunate that there is no map of the many study sites, and no picture or illustration of field sampling procedures, as well as lab instruments (especially the 'needle crushing system'). It would greatly help to have visual support for such investigations.
   ➔ Now we add the following maps and pictures of the sampling sites as well as those of gas extraction systems in the Supplement.

[Figure]

**Supplementary Figure 1.** The site locations of the ground ice samples used in this study are marked in the map of circum-Arctic permafrost (Brown et al., 1997), yedoma distributions (Strauss et al., 2016), and major rivers.

[Figure]

**Supplementary Figure 2.** Photographs of ground ice outcrops at Churapcha (central Yakutia) site. Locations of the samples used in this study are indicated by yellow dotted circles.

[Figure]

**Supplementary Figure 3.** Photographs of ground ice outcrops at Cyuie (central Yakutia) sites: (a) ice wedge outcrop, (b) CYC and (c) CYB samples. Locations of the samples used in this study are indicated by yellow dotted lines.

[Figure]

**Supplementary Figure 4.** Photographs of ground ice outcrops at Zyryanka sites: (a and b) Zy-A, (c) Zy-B, and (d) Zy-F. Locations of the samples used in this study are indicated by yellow dotted lines.

[Figure]

**Supplementary Figure 5.** Photographs of ground ice outcrops at northern Alaskan sites: (a) Bluff03 and (b) Bluff06. Locations of the samples used in this study are indicated by yellow dotted boxes.

3) Several sentences contained in the results/discussion section are in fact related to methods. I mention examples in the specific comments section below. The structure of the main text should therefore be re-aligned, so that methods sentences are in the methods section.

➔ We would like to remind that our manuscript mainly focuses on experimental- and methodological aspects. Each set of tests has been designed with logical flows. Therefore, we believe that it would be easier to read in current structure than suggested by the reviewer. However, as the re-arrangement does not change any fundamental content of our manuscript, we're willing to accept this change if the Editor suggests to do so.

4) Finally, some statements and conclusions in the main text are either not accompanied by a mention to the results or figure(s) they come from, or not supported by literature reference(s).

➔ We will revise those sentences following the specific comments below. We will also re-check the main text thoroughly.

Overall, I cannot accept this manuscript for publication as is. If the authors are willing to make major revisions (general points above and specific comments below), I would be happy to review a revised version of the manuscript.

**2) SPECIFIC COMMENTS AND EDITORIAL SUGGESTIONS**

P= page number, L = line number.

P1, L29. To avoid repetition (soil): choose either « Permafrost preserves large amounts of soil carbon and nitrogen… », or « Permafrost soils preserve large amounts of carbon and nitrogen… ».

➔ The sentence is revised as suggested by the reviewer.

➔ Permafrost  preserve large amounts of soil carbon (C) and nitrogen (N) in a frozen state (e.g., Hugelius et al., 2014; Salmon et al., 2018), temporarily removing this frozen carbon  and nitrogen  from active global cycles.

P2, L30. I suggest adding 'temporarily': «… temporarily removing this frozen carbon… ».

➔ We add 'temporarily' to the sentence. Please find our response above.

P2, L30-31. (C) and (N) should be put at the beginning of the section (P1, L29), i.e. the first time that the words 'carbon' and 'nitrogen' are mentioned.

➔ We revise the sentence following the reviewer's comment. Please refer to our response above.

P2, L35. « … which in turn can trigger positive feedbacks… ».

➔ The phrase is modified as the reviewer suggested.

➔ Therefore, future projections of permafrost stability are of great interest, particularly because thawing permafrost may lead to decomposition and/or remineralization of the buried soil C and N and their abrupt emission into the atmosphere in the form of greenhouse gases (GHGs) – carbon dioxide ($CO_2$), methane ($CH_4$), and nitrous oxide ($N_2O$), which in turn can trigger positive feedbacks (e.g., Salmon et al., 2018).

P2, L38-40. This might be true for Yedoma regions (eastern Siberia, Alaska, Yukon), but not all permafrost it necessarily ice-rich. It should be specified in the paragraph, otherwise we have the impression that permafrost all over the Arctic contains 40-90% of ground ice.

➔ The concerning sentence is revised as below.

➜ However, the processes responsible for in-situ C and N remineralization and GHG production in ground ice are poorly understood, despite the fact that ground ice accounts for a substantial portion  (up to approximately 40–90% by volume) of Pleistocene ice-rich permafrost, or Yedoma (e.g., Kanevskiy et al., 2013; Jorgenson et al., 2015).

P2, L40. I suggest adding 'Pleistocene': « … volume of Pleistocene ice-rich permafrost, or Yedoma ».
➜ We add 'Pleistocene' as suggested. Please find our response above.

P2, L42. « … evidence for in-situ microbial aerobic respiration… ». Why just 'aerobic' conditions? This might be relevant for CO2 production, but CH4 and N2O are generally produced under 'anaerobic' conditions, or both oxic-anoxic.
➜ The sentence is revised to include both aerobic and anaerobic respirations.
➜ The gases trapped in ground ice allow unique insights into the origin of ground ice and evidence for in-situ microbial aerobic and anaerobic respirations ( Boereboom et al., 2013; Kim et al., 2019; Lacelle et al., 2011).

P2, L43-44. « … detailed information on in-situ biogeochemical processes responsible for GHG production… ». Which biogeochemical processes? Methanogenesis? Respiration? Other processes?
➜ The cited literatures – Boereboom et al. (2013) and Kim et al. (2019), attributed the elevated mixing ratios of $CH_4$ and $N_2O$ to in-situ methanogenesis, nitrification, and denitrification. We note this in the sentence.
➜ Among others, the GHGs in ground ice may provide detailed information on in-situ biogeochemical processes responsible for GHG production (i.e., methanogenesis, nitrification, and denitrification) (e.g., Boereboom et al., 2013; Kim et al., 2019).

P3, L58-59. «… because ice wedges are one of the most abundant morphological features… ».
➜ Corrected.
➜ Ice-wedge samples from Alaskan and Siberian permafrost were used because ice wedges are  one of the most abundant morphological features of massive ground ice, consisting of approximately 5 to 50% by volume of the upper permafrost (Kanevskiy et al., 2013; Jorgenson et al., 2015).

P3, L73-79. For the reader not familiar with the study sites and ground-ice sampling protocols in permafrost landscapes, I strongly suggest adding 1) a map of the study sites (Siberia and Alaska); 2) pictures of an outcrop and sample collection (drilling). This way, the reader would have a much better idea of what the samples and sites look like.
➜ We add the maps showing the sampling sites along with permafrost and yedoma distributions as well as the photographs of the outcrops where our samples were taken in the Supplement. Please find our response to the general comment above and the Supplement material of the revised manuscript below.

P4, L80-97 and L88-95. Again, all these descriptions and distances would make much more sense if they were accompanied by a map (with sampling sites labeled on the map).
➜ Please find the new maps we added above.

P4, L87. «… on the first terrace of the river… ». Do the authors mean the younger (i.e. lower) terrace?
➜ Yes, this is the lowest terrace. The sentence is revised accordingly.
➜ Most of the outcrops that were sampled for ground ice were on the first (lowest) terrace of the river.

P4, L96. « The ice-wedge ice… ». This phrase is weird. Suggestions: « The ice from ice wedges is different from polar ice cores, in that… », or « Wedge ice is different…».

➔ Here we disagree. We think the "ice-wedge ice" represents "the ice from ice wedges" in a more concise way. We prefer "ice-wedge ice" to just "wedge ice" in order to better specify because there are wedges from different origins (e.g., sand wedges).

P5, L107. «… 8~13 g of ice sample were crushed… ».

➔ Corrected.

➔ In brief, 8~13 g of ice sample were crushed in a cold vacuum chamber (extraction chamber).

P5, L121-123. Which year for this modern air sample? Please specify.

➔ The modern air samples used as standard in this study were collected in November of 2016. We specify it in the sentence.

➔ …and a modern air sample from a surface firn at Styx Glacier, Antarctica (in November of 2016), which was calibrated as $1758.6 \pm 0.6$ ppb $CH_4$ and $324.7 \pm 0.3$ ppb $N_2O$ by the National Oceanic and Atmospheric Administration (NOAA).

P7, L175. I suggest 'thoroughly' or 'vigorously' instead of 'well'. («… shaken flasks were shaken thoroughly/vigorously… »

➔ Revised as suggested.

➔ After the control-wet extractions were complete, the sample flasks were shaken thoroughly  and the meltwater samples were each poured into a 50 mL conical tube.

P8, L187-189. What is meant by this statement about the heterogeneous distribution of samples? How is it shown on Fig. 1?

➔ The gas mixing ratios in the neighboring ice pieces from an ice wedge are highly variable. Our previous work (Kim et al., 2019) showed the centimeter-scale variability of the gas mixing ratios in ice wedges. We change the relavant sentence as follows:

➔ We note that the heterogenous distribution of gas mixing ratios of in centimeter scales (Kim et al., 2019) may not have been completely smoothed out by our sub-sample selection, although we randomly chose 8-12 ice cubes for each measurment.

P9, L195-196. This is a busy figure, see comments below (section 3) FIGURES). Some elements could be removed to enhance clarity.

➔ We revise Figure 1. Please refer to our response to 3) FIGURES AND TABLES below.

P10, L208-211. This reads more like methods, not a results and discussion section.

➔ Please find our response to the General Comment #3.

P10, L211-212. How about the Eastern Siberia samples? (triangles in Fig1) Did they also « not show significant differences » between the two sets of tests?

➔ The complementary tests using BES were carried out only with Central Yakutian- and Alaskan samples. The results are plotted in Figure A3 in Appendix, rather than Figure 1.

P11, L224. «… polar ice core samples… » (remove the 2nd 'ice')

➔ We delete the repetition of 'ice'.

➔ The gas extraction efficiency of the SNU needle crusher system has been reported as ~80–90% for polar ice core  samples (Shin, 2014).

P11, L230-237. This reads more like a paragraph about methods.

➔ Again, we hesitate to do so for the reason we mentioned before. Please find our response to the General Comment #3.

P11, L236-237. This information would be better displayed and more appealing in a figure.

➔ Here we hesitate to replace Table 1 with a figure. The reason why we use a table here instead of a figure is the large ranges of $CH_4$ and $N_2O$ mixing ratios that make figures less readable. Below we made a figure (grouped bar graphs) as an example. Since there are large variations among the samples, some of the data can hardly be seen, particularly for the samples having lower mixing ratios than others. Thus, we prefer to keep the Table 1, but we'll follow the Editor's decision.

P12, L250-252. This statement is based on what result? Can we see this displayed somewhere in a figure/table? If yes, please refer to it in the text.

➔ The comparisons with the dry soil content are shown both in Figures 1 and A3. We note it in the sentence.

➔ When compared with the dry soil content measured from the sub-samples used for wet extraction, no relationship was observed between the dry soil content and the extraction efficiency (Figures 1 and A3).

[Figure]

P12, L255-260. For the reader not familiar with the needle crushing system, a picture or a sketch of what the apparatus looks like might help. The sentences in this paragraph would be more easily understood.

➔ The detailed descriptions and schematic diagram of the needle crushing system used in this study can be found elsewhere in Ahn et al. (2009) or Shin (2014). However, for convenience of the reader, we add the photographs of the needle crushing system in the Supplement.

P13, L280-282. Please refer to results (figure or table) to support this statement.

➔ This sentence is our interpretation of the results from comparisons between hit5 and hit100 extractions, listed in Table 1. We note this at the end of the sentence.

➔ In the meanwhile, in the Bluff and Zyryanka samples, the hit5 results reflect the mixing ratios of the gases from the crushed portions, regardless of their origin: bubbles, particle adsorption, or microsites in aggregates (Table 1).

[Figure]

**Supplementary Figure 7?.** Schematic diagram of the needle-crusher method together with enlarged photographs of crushing needles (left top), and extraction chamber (left bottom).

P13, L288-290. Please support this statement by relevant references. In fact, the simple association N2O=oxic / CH4=anoxic is not entirely and always true. For example, N2O production has been recorded under both oxic and anoxic conditions (Gil et al. 2017; *Global Biogeochemical Cycles*), as well as CH4 production from oxic waters (Grossart et al. 2011; *PNAS*). It depends on several parameters, including local hydrology (e.g., water-logged soils). This should be acknowledged in the text.

➔ We revise the sentence as below:

➔ This can probably be explained by the fact that the $N_2O$ mixing ratio is not necessarily higher in soil-rich ice because $N_2O$ is an intermediate product of denitrification and nitrification, while $CH_4$ is produced as the final product of methanogenesis.

P13, L292 to P14, L298. This statement is highly speculative. Unless I missed something, this was not tested for real in this study. This paragraph should be supported by real data or removed.

➔ Deleted.

P14, L302-303. Again, this is methodology, not results/discussion.

➔ We believe that current structure is easier to read for the same reason above. Please find our response to the General Comment #3.

P16, L320-330. Again: methodology, not results/discussion.

➜ Again, we retain the current structure for the same reason we answered to the General Comment #3.

P17, L350-352. This is indeed interesting. Do we know why N2O appears to be more extractable (or less present in the residual adsorbed phase) than CH4, at least based on the wet extraction technique? Was this already observed elsewhere and reported in the literature?

➜ Currently we speculate that higher solubility (to water) of $N_2O$ could make the adsorbed $N_2O$ more extractable by wet extraction, compared to $CH_4$. To our best knowledge, this is reported for the first time by our manuscript. We added a sentence mentioning this as below:

➜ These results imply that most of the $N_2O$ in ice wedges is extracted by three melting-refreezing cycles, such that only a small amount of $N_2O$ is left adsorbed or entrapped in ice-wedge soils. It might be attributed to the high solubility of $N_2O$ to water compared to $CH_4$.

P18, L375. «… easy to extract… »

➜ The typo is corrected.

P19, L397. « Our findings indicate that ... »

➜ Corrected.

**3) FIGURES AND TABLES**

Figure 1, P9.
- ➔ This is a pretty busy figure. We don't necessarily need the 3 legends (identical) in the middle. By removing them, more space could be created to enlarge the graphs a bit, because for now they are quite small. Also: what is the purpose of the insets (a-b, e-f)?
- ➔ The purpose of the insets is to better show the data points in the low ranges. The Figure 1 is now modified as below.

[Figure]

a) I don't get the thing about the error bars (in blue). Are these 5x, 100x or 500x larger or smaller in 'real life' than displayed on the graphs? Not clear.
  ➔ The data uncertainties are too small to be plotted. Thus, the blue error bars in Figure 1 are magnified by 5x, 100, and 500x. We will add words in the figure caption so that we better clarify the numbers.

b) Explain what does 'hit5' mean.
  ➔ The meaning of 'hit5' and 'hit100' are already explained in the main text of Section 3.3. However, we add a sentence explaining 'hit5' in the figure caption for better readability.
  ➔ **Figure 1.** Comparison of $CH_4$ and $N_2O$ mixing ratios and contents obtained by different extraction methods. Shown are scatter plots between wet- and dry (hit5) extraction results of $CH_4$ (a and b) and $N_2O$ (c and d), and between control- and biocide-treated wet extraction results for $CH_4$ (e) and $N_2O$ (f). The 'hit5' denotes the dry extraction with five times hitting (see Section 3.3). Left panels (a, c, and e) and (f) present in mixing ratios of gas in bubbles, while right (b) and (d) panels in moles of gas in a unit mass of ice (gas content). The sampling locations are indicated by different symbols. The color of each data point indicates the dry soil weight in the subsamples used in control wet extraction. The 1-sigma uncertainties of the mixing ratios (a, c, e, and f) are denoted as blue error bars (see Appendix). The error bars are not visible where the error bars are smaller than markers. The grey dashed lines are 1:1 reference line. Note that the units of the axes of the insets in (e) and (f) are identical to the original plots. The p-value of two-sided Students' t-test of each comparison is denoted at the top of each plot.

Table 1, P15.
  a) It is not explained why the hit100/hit5 ratios for gas content (6th column) are much lower for most of the central Yakutia samples (Cyuie), compared to the other sites? This is indeed interesting, but why? Less soil aggregates in ice-wedge samples from this site, so relatively more bubbles and thus more extracted gas?
    ➔ Indeed, we already addressed this in the main text of Section 3.3. The low hit100/hit5 ratios of gas content in Cyuie samples are attributed to the easier-crushing characteristics of the Cyuie samples compared to the others, so that much of the enclosed gases are extracted by hit5 extraction.

Figure 2, P18.
  a) Where do the samples come from? CYC-02-B and CYC-03-C likely refer to Central Yakutia (Cyuie), but what about the other samples (C-04, C-30, C-10, C-12)? Please specify somewhere.
    ➔ The samples of 'C-##' come from Churapcha site. We modified the figure caption to specify the sample origin.
    ➔ **Figure 2. Comparison of wet-extracted gas and residual gas for $CH_4$ and $N_2O$ mixing ratios (a and b) and contents (c and d).** The residual gas was extracted from the dry extraction method using the wet-degassed ice samples. The light green bars show the results of initial wet extraction, and the blue and red bars indicate the dry extraction of wet-degassed ice with 20- and 60-times hitting, respectively. The Cyuie samples are denoted as 'CYC', while 'C' indicates the Churapcha samples.

---

## Author Comment (AC2) · 14 Jan 2020

**We would like to thank the two anonymous reviewers and the Editor for their careful reviews of our manuscript. Below we present our point-by-point responses to all of the comments and annotations. The original comments from the reviewers are shown in black, while our responses are presented in red. Where indicated, additions to the manuscript are shown in blue and removed passages shown in red strikethrough font.**

========================================================================================

**Comments from the reviewer #2 and our responses:**

1) GENERAL COMMENTS

The paper reports novel aspects of experimental methods of gas extraction techniques. Currently, there are no works that conducted such a comparison, which has long been needed due to huge number of gas measurements in permafrost area conducted recently. Unfortunately, there is still no unified method for carrying out gas extraction that leads to the impossibility and impropriety of comparison the results obtained by different research groups.

The authors tested conventional wet and dry gas extraction methods for ice wedges coming to conclusion that current estimates of ground-ice gas budgets are likely underestimated. They found insignificant effects of microbial activity during wet extraction and significant difference in extraction results from polar ice cores and ice wedges. Therefore, the manuscript is of big interest for scientific community and contributes to changing our scientific understanding of a subject as it is has to be for TC.

The results are presented in well-structured way and the paper is easy to read.

However, there are a few general suggestions that could improve the article:
1)       I suggest adding a map of study sites, maybe some geological sections to get a better idea of the location and structure of ice wedges. Are they all Pleistocene?

➔ We agree with the reviewer. We add a map in our Supplement (see below). The ages of the studied ice wedges have yet been analyzed, and they are beyond the scope of our manuscript.

[Figure]

**Supplementary Figure 1.** The site locations of the ground ice samples used in this study are marked in the map of circum-Arctic permafrost (Brown et al., 1997), yedoma distributions (Strauss et al., 2016), and major rivers.

2)      Besides it is really necessary to include the schemes of gas extraction procedures (both wet and dry techniques as well as the experiment on dry extraction efficiency and on residual gas contents after wet extraction.  Due to the limitation of the number of figures both 1 and 2 can be added as supplementary material.

➔ We add figures in the Supplement for readers. As we already cited in the text, the details of the extraction system and procedures are well described in Ahn et al. (2009) and Shin (2014) for dry extraction and Yang et al. (2017) and Ryu et al. (2018) for wet extraction.

[Figure]

**Supplementary Figure 6.** Schematic diagram of needle-crusher method together with enlarged photographs of crushing needles (left top), and extraction chamber (left bottom). The detailed descriptions about the SNU dry extraction system can be found elsewhere in Ahn et al. (2009) and Shin (2014).

[Figure]

**Supplementary Figure 7.** Schematic diagram of melting-refreezing (wet extraction) procedure used in this study. More details about the wet extraction line and GC systems are described in Yang et al. (2017) and Ryu et al. (2018).

3)        Since the article is devoted to the comparison of methods, it would be useful to estimate the limits of applicability of the methods and measurement errors.

➔ We agree with the reviewer's comment, but the limits of applicability and measurement errors of both methods were already described in the main text (Section 3.3 and 3.4) and Appendix, respectively.

4)        It is necessary to add initial data on gas content and $CH_4$ and $N_2O$ mixing ratios as supplementary material to prove you main result about the same effectiveness of wet and dry extraction methods. As I see now from the Table 1 there can be 2 times difference (up to 20000 ppm) for $CH_4$

➔ As we mentioned above, we will upload our original data in a public data repository (PANGAEA) once our manuscript is accepted.

➔ We provide the relative amount of extracted gas in Table 1.

So the article is of big interest but needs major revision to be accepted and I would be happy to review a revision of the paper.

2) SPECIFIC COMMENTS AND EDITORIAL SUGGESTIONS

P.2 L50-51. «…ice sample was melted in a saturated sodium chloride (NaCl) solution, in order to minimize microbial activity and gas dissolution (Cherbunina et al., 2018 and references therein)». I found no mention in the article that NaCl was used in order to minimize microbial actvity, is it really there?

➔ We correct the reference as below:

➔ Other studies conducted by Russian scientists used an on-site melting method in which a large (1–3 kg) block of ground ice sample was melted in a saturated sodium chloride (NaCl) solution, in order to minimize  gas dissolution (Arkhangelov and Novgorodova, 1991).

➔ Arkhangelov, A. A., and Novgorodova, E. V.: Genesis of massive ice at 'Ice Mountains', Yenesei River, Western Siberia, according to results of gas analyses, Permafrost Periglac. Proc., 2, 167-170, http://doi.org/10.1002/ppp.3430020210, 1991.

P.2 L72. Please specify the size of the samples, add the site map and geological sections with sampling location

➔ We add the map of sampling site as Supplement Figure 1 (please refer to our response to General Comment #1 above). We also add photos of outcrops where our samples were taken.

[Figure]

**Supplementary Figure 2.** Photographs of ground ice outcrops at Churapcha (central Yakutia) site. Locations of the samples used in this study are indicated by yellow dotted circles.

[Figure]

**Supplementary Figure 3.** Photographs of ground ice outcrops at Cyuie (central Yakutia) sites: (a) ice wedge outcrop, (b) CYC and (c) CYB samples. Locations of the samples used in this study are indicated by yellow dotted lines.

[Figure]

**Supplementary Figure 4.** Photographs of ground ice outcrops at Zyryanka sites: (a and b) Zy-A, (c) Zy-B, and (d) Zy-F. Locations of the samples used in this study are indicated by yellow dotted lines.

[Figure]

**Supplementary Figure 5.** Photographs of ground ice outcrops at northern Alaskan sites: (a) Bluff03 and (b) Bluff06. Locations of the samples used in this study are indicated by yellow dotted boxes.

P.2 L104. Please include the schemes of gas extraction procedures
➔ Please refer to our response to the General Comment #2 above.

P.2 L108. Why it is used precisely 5 times, not 100, can you reason it somehow?
➔ We have empirical knowledge that the polar ice core samples are well crushed within five times hitting by our dry extraction (needle-crusher) at SNU. Therefore, for consistency of analytical setup, we applied the identical procedure of dry extraction to ice-wedge samples to test whether the dry extraction method is applicable. In the meanwhile, the tests with 100-times hitting were designed to understand gas extraction efficiency and difference in gas mixing ratios in between easily- and hardly crushed portions of ice wedges.

P.2 L144. It is not clear where did you get the dry soil mass before the extraction. «Taking the dry soil mass of the analysed samples (0.33 g) into account, we added 24 µL of saturated HgCl2 solution (at 20°C) to the sample flasks» Were there used the data on dry soil mass from the other samples? Because later you say « Dry soil content was measured using the leftover meltwater from the control-wet extraction tests. »
➔ We obtained the dry soil mass (0.33 g) from the leftover meltwater samples of the **previous** wet extractions, which wrere carried out for comparison between dry- and wet extractions. To clarify we will revise the words in the revision.

P.12 L249. Please specify what do you mean by «ice hardness» here. As in L 249 «the extraction efficiency of the needle crusher not only depends on site characteristics, but also on the individual ice sample hardness», and later L. 251 «no relationship was observed between the dry soil content and the extraction efficiency», but L.274 « soil-rich ice has greater hardness than the soil-poor ice». I guess this is the matter of «soil aggregates» as you mention later, so the hardness in this case is defined by this parameter? Is it possible to quantify this?

➔ We observed that the samples containing large-sized soil aggregates are hardly crushed by our needle-crusher system. Since no significant relationship was found between dry soil content and the extraction efficiency, it may imply that the important parameter controlling hardness is the presence of large-sized aggregate, rather than just soil content. Unfortunately, we have no quantitative measure of size (or volume) of each soil aggregate. Further study with three-dimensional image analysis will be useful to address this.

➔ To clarify this issue, we will reword the sentence L.274 "soil-rich ice has greater hardness than the soil-poor ice" like as follows:

➔ Thus, the hit5 $CH_4$ mixing ratios of the Cyuie samples may more reflect the gas mixing ratios in bubbles, while the hit100 results reflect more of the contribution from gas adsorbed on soil and trapped within soil aggregates than the hit5 results because the ice sample containing larger-sized aggregates has greater hardness than those with smaller aggregates or fine particles.

P.12 L255. Please specify the size range for «This is because the large-sized uncrushed soil aggregates or particles may have prohibited the needle crusher from crushing the small-sized ice flakes or grains». As the presence of the aggregates is one of the main limits to use the technique, is it possible to make at least a rough estimate of the amount of gas that can remain there?

➔ Although we have no quantitative measure of size of soil aggregate, the soil aggregates in the studied ice wedges are observable by naked eyes as they have clear contrast of darkness when back-lighted. Empirically, the size of observable aggregates is in millimeter-scale, while some of them reached a centimeter.

➔ However, it is not possible to estimate the amount of gas remained in the uncrushed portion (mostly Zyryanka and Bluff samples), because it is unknown how much gas is entrapped there. On the other hand, in case for the easily-crushed samples (i.e. Cyuie samples), the hit100/hit5 ratio of gas content could be used to estimate the gas amount in soil aggregates. However, we cannot recommend this estimation because a certain portion of gas in soil aggregates could also been extracted by hit5 procedures, and we have no information on how much of soil aggregates are crushed or uncrushed after the hit5 procedures.

P.12 L259. «Therefore, we do not recommend using a needle crusher system to measure gas contents in ice-wedge samples». Can you estimate the efficiency of the method in % in the same way it has been done for polar ice core ice samples (80–90%) (Shin, 2014)? As I see from the Table 1 the procedure «Hit5+Hit100» in most cases allows to extract more gas then the wet method even if uncrushed aggregates still occur. Can you recommend using dry extraction method in this modification?

➔ We can't. To measure gas content precisely, it requires a near-perfect gas extraction from ice wedge samples. As the reviewer pointed out, it is true that the amount of gas extracted from both hit5 and hit100 procedures is generally higher than wet extraction. However, we hesitate to make a general statement because the uncrushed soil aggregates may still exist even after hit100 extraction, depending on sampling locations and individual samples.

P.12 L272. Since you talk about gas in bubbles here : «the hit5 $CH_4$ mixing ratios of the Cyuie samples may more reflect the gas mixing ratios in bubbles, while the hit100 results reflect more of the contribution from gas adsorbed on soil and trapped within soil aggregates than the hit5 results» and further, may be it would be useful to get the data on ice porosity to compare with the results of extracted volume of gas since the volume of gas normalized to layer pressure approximately corresponds to porosity.

➔ This is a great idea indeed. However, the relationship between the porosity and the normalized volume would work only when gas extraction efficiency is near 100%, or

near constant. The gas extraction efficiency of the ice wedge is highly variable and difficult to be precisely measured, limiting estmation of the porosity.

P.16 L320. Please explain if I understand correctly the next paragraph:

«To examine how well the gas is extracted by wet extraction, we applied the dry extraction method to refrozen ice-wedge samples after wet extraction. We first prepared degassed ice-wedge samples that had undergone repetitive wet extractions (wet-degassed ice hereafter). Once the wet extraction experiments were completed, we repeated two cycles of melting-refreezing and evacuation procedures to degas the ice melt. After degassing by a total of three cycles of wet extraction and evacuation, the outermost surfaces (~2 mm) of the wet degassed ice were trimmed away in the walk-in freezer at SNU on the morning of experiments. The wet-degassed ice was then inserted into the needle crusher and the crusher chamber was evacuated. A specific amount of standard air was injected. Then, the wet-degassed ice samples were hit 20 or 60 times by the needle crusher.»

After the first freezing-melting cycle the sample gas in the headspace of the flask was collected and gas content, $CH_4$ and $N_2O$ ratio was measured. Then the two cycles were conducted. (and my question here is what happened to the gas in the flask-was it collected and measured or just evacuated) and next step was measuring the gas content in degassed ice through needle-crusher procedure.

➔ The reviewer understands correctly. Regarding the question, we didn't collect the gas extracted by 2nd and 3rd cycles of melting-refreezing procedures.

P.16 L328. Explain please why were such parameters chosen if in the previous dry extraction procedure you used 5 and 100 hits: « Then, the wet-degassed ice samples were hit 20 or 60 times by the needle crusher»

➔ The main goal of the tests with the wet-degassed ice samples is to know if there is gas remained after three-cycles of wet extraction. We chose 20-times hitting instead of 5 times because significant amount of gas was already extracted from the 3-cycled wet extractions. We also chose 60-times hitting to see if there is a significant difference in amount of the extracted gas between the 20- and 60-times hittings.

P.16 L331. The tests using the wet-degassed ice show an additional gas extraction of 43 to 88% of the amount of gas extracted during the initial wet extraction. I suggest to add this information to the conclusions as it is of big significance as well as if I get it right the best way of degassing the sample according to your manuscript is to combine three cycles of wet extraction with dry extraction for the residual gas. I think this has to be one of the main conclusion.

➔ The more number of wet and dry extractions, the better gas extraction efficiency. We think we better leave the conclusions as they are because we cannot specify the best combination in numbers of the two extraction methods.

P.19 L391. Please specify what do you mean by «relatively soft ice wedges».

➔ Here we refer to the ice wedges that are more easily crushed than others by the hit5 procedure. To specify this, we will revise the sentence like below:

➔ In the meantime, we propose that both existing techniques may be suitable for gas mixing ratio measurements for bubbles in relatively soft ice wedges (i.e., easily crushed ice wedges by a hit5 extraction, e.g., Cyuie ice wedges in this study).

P.19 L 392  It seems to me that you have very good results of applying the method of three times wet extractions + residual gas extracted by a needle crusher for $N_2O$ and I don't get why there is in conclusions «Exceptionally, the N2O content in ice wedges may be measured by using repeated wet extractions, but this is not the case for determining the N2O mixing ratio»

➔ As the reviewer pointed out, our results indicate that the repeated melting-refreezing procedures extract most of the $N_2O$ from ice wedges. However, we cannot guarantee the $N_2O$ mixing ratio because the relative extraction efficiency for each gas species may be variable. To clarify, we will add words in the main text.

3) FIGURES AND TABLES

Table 1. Add a column of dry soil content as in table 2

➔ We revise the Table 1 as suggested (see next page).

The table with the data used for Fig.1 need to be added to get the difference between wet and dry method results.

➔ We will add a table for the data used for Fig. 1 in Supplement.

| Site Location | Sample | soil content | gas content | | | | $CH_4$ mixing ratio | | | | $N_2O$ mixing ratio | | | |
|---|---|---|---|---|---|---|---|---|---|---|---|---|---|---|
| | | | Wet control | Dry hit5 | Dry hit100 | hit100/hit5 | Wet control | Dry hit5 | Dry hit100 | hit100/hit5 | Wet control | Dry hit5 | Dry hit100 | hit100/hit5 |
| | | wt. % | ml/kg | ml/kg | ml/kg | | ppm | ppm | ppm | | ppm | ppm | ppm | |
| Zyryanka, Northeastern Siberia | Zy-A-W1-D | 0.155 | 20.2 | 13.1 | 6.3 | 0.48 | 6138 | 3713 | 2721 | 0.7329 | 11.37 | 9.10 | 10.15 | 1.12 |
| | Zy-F-1 | 0.618 | 13.5 | 8.1 | 3.4 | 0.42 | 1080 | 655.6 | 173.5 | 0.2646 | 1.57 | 2.81 | 2.65 | 0.942 |
| | Zy-A-W1-Low | 0.049 | 30.6 | 27.8 | 8.0 | 0.29 | 4309 | 5073 | 4818 | 0.9497 | 2.07 | 0.69 | 2.02 | 2.9 |
| | Zy-B-Low-B | 0.107 | 29.1 | 23.9 | 10.0 | 0.418 | 18030 | 21010 | 35290 | 1.680 | 5.37 | 5.32 | 15.36 | 2.89 |
| Northern Alaska | Bluff03-IW1 | 2.07 | 13.2 | 12.2 | 2.6 | 0.21 | 44160 | 25230 | 12240 | 0.4851 | 5.58 | 2.36 | 4.93 | 2.09 |
| | Bluff06-B3 | 0.078 | 20.1 | 20.9 | 5.6 | 0.27 | 558.7 | 164.2 | 219.5 | 1.337 | 3.74 | 18.78 | 30.14 | 1.605 |
| Cyuie, Central Yakutia | CYC-01-B | 0.252 | 18.0 | 21.7 | 7.1 | 0.33 | 18.0 | 18.3 | 25.4 | 1.39 | 1.55 | 1.60 | 2.59 | 1.62 |
| | CYB-04-C | 0.498 | 20.9 | 30.7 | 1.5 | 0.049 | 20.2 | 48.4 | 165.6 | 3.42 | 0.71 | 0.65 | 2.96 | 4.5 |
| | CYB-03-A | 0.420 | 19.7 | 23.7 | 1.0 | 0.041 | 20.5 | 21.5 | 67.1 | 3.12 | 0.91 | 1.01 | 1.06 | 1.05 |
| | CYB-02-A | 0.403 | 32.0 | 25.5 | 1.9 | 0.073 | 29.1 | 18.7 | 159.8 | 8.55 | 1.00 | 0.58 | 3.19 | 5.5 |
| | CYC-03-B | 0.830 | 22.6 | 15.7 | 3.3 | 0.21 | 20.3 | 13.9 | 94.5 | 6.80 | 1.40 | 0.65 | 1.08 | 1.7 |

---

## Author Response (AR1)

**We would like to thank the reviewers and the Editor for their careful review of our manuscript. Our point-by-point responses to all the comments and annotations are presented below. The comments from the reviewers are shown in black, and our responses in red. Where indicated, additions to the manuscript are shown in blue and deleted text is indicated by red strikethrough, and the pages and line numbers are based on our revised (non-track) manuscript.**

===============================================================================

**Comments from the reviewer #1 and our responses:**

**1) GENERAL COMMENTS**

This methodological, brief communication paper reports on a comparison of different techniques (wet *vs.* dry extraction, with or without biocide to test microbial contamination) to extract gas (CH4 and N2O) from ice wedges of Alaska and Siberia. The authors report that tested methods yield good results for the easily extractable gas fraction (bubbles), but this is not so convincing for the adsorbed phase or gas contained within soil aggregates. One of the main conclusions, therefore, is that current estimates of ground-ice gas budgets are likely underestimated, as a fraction of produced gases are not taken into account. For me, this is the main take-home message.

It appears as an interesting short paper, although the methodology used is not in my immediate field of expertise. To my knowledge, this manuscript does not have major flaws that should ultimately prohibit its publication. It is generally well written and easy to read. I have however a few points to mention that preclude acceptance for publication as is:

1) I am not convinced, for now, of the general, broad-audience impact of the manuscript. Does it really « *report new developments, significant advances, and novel aspects of experimental and theoretical methods and techniques which are relevant for scientific investigations within the journal scope* »? (https://www.the-cryosphere.net/about/manuscript_types.html). The authors have not convinced me that this work is new, innovative or represent a major advancement that is relevant to the community at large. They rather suggest that a future, novel extraction method might provide better results. I am also looking forward to that. This work might be useful for a small specialized group, however. Furthermore, the conclusion about underestimation of current gas budgets in ice-wedge terrains is itself interesting and timely.

➔ We believe that our manuscript has broad implications for the large community studying on permafrost-climate interactions. Permafrost thawing is a major potential global warming threat, which is expected to input large amount of greenhouse gas (GHG) into the atmosphere. Thus, it is important to quantify the permafrost GHG budget is important for better projection of climate change. There are many works on methane ($CH_4$) and nitrous oxide ($N_2O$) concentrations either in permafrost or ground ice. However, there is currently no consensus on gas extraction methods, and no method has been tested properly. To the best of our knowledge, this study is the first to test different gas extraction methods to understand the applicability and limitations of both techniques. We believe that our study makes a significant contribution to the literature that while existing methods allow gas extraction from the soft parts of ice, gas adsorbed by or trapped in soil particles may not be extracted, thus affecting the measurement of GHG contents and their mixing ratios. In addition, we show that the microbial activities have a negligible effect on the wet extraction results. Although our manuscript does not deal with a *<new development>*, we believe that these findings provide *<significant advances>* and *<novel aspects of experimental methods>*. The development of a new technique is far beyond the scope of this manuscript.

2) I understand that this is a brief communication and that the number of figures/tables is limited. However, it is really unfortunate that there is no map of the many study sites, and no picture or illustration of field sampling procedures, as well as lab instruments (especially the 'needle crushing system'). It would greatly help to have visual support for such investigations.

➔ We add the following maps and images of the sampling sites as well as those of the gas extraction systems, in the Supplement.

[Figure]

**Supplementary Figure 1.** The site locations of the ground ice samples used in this study are marked in the map of circum-Arctic permafrost (Brown et al., 2002), yedoma distributions (Strauss et al., 2016), and major rivers.

[Figure]

**Supplementary Figure 2.** Photographs of ground ice outcrops at Churapcha (central Yakutia) site. Locations of the samples used in this study are indicated by yellow dotted circles.

[Figure]

**Supplementary Figure 3.** Photographs of ground ice outcrops at Cyuie (central Yakutia) sites: (a) ice wedge outcrop, (b) CYC and (c) CYB samples. Locations of the samples used in this study are indicated by yellow dotted lines.

[Figure]

**Supplementary Figure 4.** Photographs of ground ice outcrops at Zyryanka sites: (a and b) Zy-A, (c) Zy-B, and (d) Zy-F. Locations of the samples used in this study are indicated by yellow dotted lines.

[Figure]

**Supplementary Figure 5.** Photographs of ground ice outcrops at northern Alaskan sites: (a) Bluff03 and (b) Bluff06. Locations of the samples used in this study are indicated by yellow dotted boxes.

3) Several sentences contained in the results/discussion section are in fact related to methods. I mention examples in the specific comments section below. The structure of the main text should therefore be re-aligned, so that methods sentences are in the methods section.

➔ Our study mainly focuses on the experimental- and methodological aspects. Each set of tests was designed with logical flow. Therefore, we believe it would be easier to follow the study in the current structure than the one by the reviewer. However, as the re-arrangement does not change any fundamental content of our manuscript, we are willing to make this change if the Editor suggests so.

4) Finally, some statements and conclusions in the main text are either not accompanied by a mention to the results or figure(s) they come from, or not supported by literature reference(s).

➔ We revise these sentences as per the specific comments below. We also re-check the main text thoroughly.

Overall, I cannot accept this manuscript for publication as is. If the authors are willing to make major revisions (general points above and specific comments below), I would be happy to review a revised version of the manuscript.

**2) SPECIFIC COMMENTS AND EDITORIAL SUGGESTIONS**

P= page number, L = line number.

P1, L29. To avoid repetition (soil): choose either « Permafrost preserves large amounts of soil carbon and nitrogen… », or « Permafrost soils preserve large amounts of carbon and nitrogen… ».

➔ The sentence is revised as per the reviewer's suggestions.

➔ (P1. L29.) Permafrost  preserves large amounts of soil carbon (C) and nitrogen (N) in a frozen state (e.g., Hugelius et al., 2014; Salmon et al., 2018), temporarily removing this frozen carbon  and nitrogen  from active global cycles.

P2, L30. I suggest adding 'temporarily': «… temporarily removing this frozen carbon… ».

➔ We add 'temporarily' to the sentence. Please find responses above.

P2, L30-31. (C) and (N) should be put at the beginning of the section (P1, L29), i.e. the first time that the words 'carbon' and 'nitrogen' are mentioned.

➔ We revise the sentence following the reviewer's comment. Please refer to our above response.

P2, L35. « … which in turn can trigger positive feedbacks… ».

➔ The phrase is modified as per the reviewer's suggestion.

➔ (P2. L31-35.) Therefore, future projections of permafrost stability are of great interest, particularly because thawing permafrost may lead to decomposition and/or remineralization of the buried soil C and N and their abrupt emission into the atmosphere in the form of greenhouse gases (GHGs) – carbon dioxide ($CO_2$), methane ($CH_4$), and nitrous oxide ($N_2O$), which in turn can trigger positive feedbacks (e.g., Salmon et al., 2018).

P2, L38-40. This might be true for Yedoma regions (eastern Siberia, Alaska, Yukon), but not all permafrost it necessarily ice-rich. It should be specified in the paragraph, otherwise we have the impression that permafrost all over the Arctic contains 40-90% of ground ice.

➔ This sentence is revised as follows.

➔ (P2. L37-40.) However, the processes responsible for in-situ C and N remineralization and GHG production in ground ice are poorly understood, despite the fact that ground ice accounts for a substantial portion  (up to approximately 40–90% by volume) of Pleistocene ice-rich permafrost, or Yedoma (e.g., Kanevskiy et al., 2013; Jorgenson et al., 2015).

P2, L40. I suggest adding 'Pleistocene': « … volume of Pleistocene ice-rich permafrost, or Yedoma ».

➔ We add 'Pleistocene' as suggested. Please find our response above.

P2, L42. « … evidence for in-situ microbial aerobic respiration… ». Why just 'aerobic' conditions? This might be relevant for CO2 production, but CH4 and N2O are generally produced under 'anaerobic' conditions, or both oxic-anoxic.

➔ This sentence is revised to include both aerobic and anaerobic respirations.

➔ (P2. L41-43.) The gases trapped in ground ice allow unique insights into the origin of ground ice and evidence for in-situ microbial aerobic and anaerobic respirations ( Boereboom et al., 2013; Kim et al., 2019; Lacelle et al., 2011).

P2, L43-44. « … detailed information on in-situ biogeochemical processes responsible for GHG production… ». Which biogeochemical processes? Methanogenesis? Respiration? Other processes?

➔ The cited literatures – Boereboom et al. (2013) and Kim et al. (2019), attributed the elevated mixing ratios of $CH_4$ and $N_2O$ to in-situ methanogenesis, nitrification, and denitrification. We note this in the sentence.

➔ (P2. L43-46.) Among others, the GHGs in ground ice may provide detailed information on in-situ biogeochemical processes responsible for GHG production (i.e., methanogenesis, nitrification, and denitrification) (e.g., Boereboom et al., 2013; Kim et al., 2019).

P3, L58-59. «… because ice wedges are one of the most abundant morphological features… ».

➔ Corrected.

➔ (P3. L59-62.) Ice-wedge samples from Alaskan and Siberian permafrost were used because ice wedges are  one of the most abundant morphological features of massive ground ice, consisting of approximately 5 to 50% by volume of the upper permafrost (Kanevskiy et al., 2013; Jorgenson et al., 2015).

P3, L73-79. For the reader not familiar with the study sites and ground-ice sampling protocols in permafrost landscapes, I strongly suggest adding 1) a map of the study sites (Siberia and Alaska); 2) pictures of an outcrop and sample collection (drilling). This way, the reader would have a much better idea of what the samples and sites look like.

➔ We add the maps showing the sampling sites along with permafrost and yedoma distributions as well as images of the outcrops where our samples were taken in the Supplement. Please find our response to the general comment above and the Supplement material of the revised manuscript below.

P4, L80-97 and L88-95. Again, all these descriptions and distances would make much more sense if they were accompanied by a map (with sampling sites labeled on the map).

➔ We have shown the new maps above.

P4, L87. «… on the first terrace of the river… ». Do the authors mean the younger (i.e. lower) terrace?

➔ Yes, this is the lowest terrace. The sentence is revised accordingly.

➔ (P4. L88-89.) Most of the outcrops that were sampled for ground ice were on the first (lowest) terrace of the river.

P4, L96. « The ice-wedge ice… ». This phrase is weird. Suggestions: « The ice from ice wedges is different from polar ice cores, in that… », or « Wedge ice is different…».

➔ Here we disagree. We believe the "ice-wedge ice" represents "the ice from ice wedges" in a more concise way. We prefer "ice-wedge ice" to just "wedge ice" for better specificity because there are wedges from different origins (e.g., sand wedges).

P5, L107. «… 8~13 g of ice sample were crushed… ».

➔ Corrected.

➔ (P5. L108-109.) In brief, 8~13 g of ice sample were crushed in a cold vacuum chamber (extraction chamber).

P5, L121-123. Which year for this modern air sample? Please specify.

➔ The modern air samples used as standard in this study were collected in November of 2016. We revise the following sentence accordingly.

➔ (P5. L122-125.) …and a modern air sample from a surface firn at Styx Glacier, Antarctica (obtained in November 2016), which was calibrated as 1758.6 ± 0.6 ppb $CH_4$ and 324.7 ± 0.3 ppb $N_2O$ by the National Oceanic and Atmospheric Administration (NOAA).

P7, L175. I suggest 'thoroughly' or 'vigorously' instead of 'well'. («… shaken flasks were shaken thoroughly/vigorously… »

➔ Revised as suggested.

➔ (P7. L177-178.) After the control-wet extractions were complete, the sample flasks were shaken thoroughly  and the meltwater samples were each poured into a 50 mL conical tube.

P8, L187-189. What is meant by this statement about the heterogeneous distribution of samples? How is it shown on Fig. 1?

➔ The gas mixing ratios obtained from neighboring ice-wedge pieces were highly variable. Our previous work (Kim et al., 2019) showed the centimetre-scale variability of the gas mixing ratios in ice wedges. We change the relevant sentence as follows:

➔ (P8. L190-193.) We note that the heterogenous distribution of gas mixing ratios of in centimeter scales (Kim et al., 2019) may not have been completely smoothed out by our sub-sample selection, although we randomly chose 8-12 ice cubes for each measurement.

P9, L195-196. This is a busy figure, see comments below (section 3) FIGURES). Some elements could be removed to enhance clarity.

➔ We revise Figure 1. Please refer to our response to 3) FIGURES AND TABLES below.

P10, L208-211. This reads more like methods, not a results and discussion section.

➔ Please find our response to the General Comment #3.

P10, L211-212. How about the Eastern Siberia samples? (triangles in Fig1) Did they also « not show significant differences » between the two sets of tests?

➔ The complementary tests using BES were carried out only with Central Yakutian- and Alaskan samples. The results are plotted in Figure A3 in Appendix, rather than Figure 1.

P11, L224. «… polar ice core samples… » (remove the 2nd 'ice')
   ➔ We delete the repetition of 'ice'.
   ➔ (P11. L227-228.) The gas extraction efficiency of the SNU needle crusher system has been reported as ~80–90% for polar ice core  samples (Shin, 2014).

P11, L230-237. This reads more like a paragraph about methods.
   ➔ Again, we hesitate to do so for the reason we mentioned before. Please find our response to the General Comment #3.

[Figure]

P11, L236-237. This information would be better displayed and more appealing in a figure.
   ➔ Here we hesitate to replace Table 1 with a figure. We used a table originally because of the large ranges of $CH_4$ and $N_2O$ mixing ratios, which make figures less readable. Below we have made a figure (grouped bar graphs) as an example. Because there are large variations among the samples, some data are difficult to see, particularly for samples with lower mixing ratios. Thus, we prefer use Table 1; however, we will follow the Editor's decision.

P12, L250-252. This statement is based on what result? Can we see this displayed somewhere in a figure/table? If yes, please refer to it in the text.
   ➔ We add that the comparisons with the dry soil content are shown both in Figures 1 and A3.
   ➔ (P12. L254-256.) When compared with the dry soil content measured from the sub-samples used for wet extraction, no relationship was observed between the dry soil content and the extraction efficiency (Figures 1 and A3).

P12, L255-260. For the reader not familiar with the needle crushing system, a picture or a sketch of what the apparatus looks like might help. The sentences in this paragraph would be more easily understood.

➔ Detailed descriptions and schematic diagram of the needle crushing system can be found in Ahn et al. (2009) and Shin (2014). We add images of the needle crushing system to the Supplement.

[Figure]

**Supplementary Figure 6.** Schematic diagram of the needle-crusher method together with enlarged photographs of crushing needles (left top), and extraction chamber (left bottom).

P13, L280-282. Please refer to results (figure or table) to support this statement.

➔ This sentence is our interpretation of the comparisons of hit5 with hit100 extractions, listed in Table 1. We note this at the end of the sentence.

➔ (P13. L284-287.) In the meanwhile, in the Bluff and Zyryanka samples, the hit5 results reflect the mixing ratios of the gases from the crushed portions, regardless of their origin: bubbles, particle adsorption, or microsites in aggregates (Table 1).

P13, L288-290. Please support this statement by relevant references. In fact, the simple association N2O=oxic / CH4=anoxic is not entirely and always true. For example, N2O production has been recorded under both oxic and anoxic conditions (Gil et al. 2017; *Global Biogeochemical Cycles*), as well as CH4 production from oxic waters (Grossart et al. 2011; *PNAS*). It depends on several parameters, including local hydrology (e.g., water-logged soils). This should be acknowledged in the text.

➔ We revise the sentence as follows:

➔ (P13. L292-295.) This can probably be explained by the fact that the $N_2O$ mixing ratio is not necessarily higher in soil-rich ice because $N_2O$ is an intermediate product of denitrification and nitrification, while $CH_4$ is produced as the final product of methanogenesis.

P13, L292 to P14, L298. This statement is highly speculative. Unless I missed something, this was not tested for real in this study. This paragraph should be supported by real data or removed.

➔ Deleted.

P14, L302-303. Again, this is methodology, not results/discussion.

➔ We believe that the current structure is easier to read for the same reason discussed above. Please find our response to General Comment #3.

P16, L320-330. Again: methodology, not results/discussion.
➔ We wish to retain the current structure for the same reason mentioned in the response to General Comment #3.

P17, L350-352. This is indeed interesting. Do we know why N2O appears to be more extractable (or less present in the residual adsorbed phase) than CH4, at least based on the wet extraction technique? Was this already observed elsewhere and reported in the literature?
➔ Currently we speculate that higher solubility (to water) of $N_2O$ could make the adsorbed $N_2O$ more extractable by wet extraction, compared with $CH_4$. To the best of our knowledge, this is reported for the first time by our manuscript. Thus, we add a sentence mentioning this as below:
➔ (P17. L355-359.) These results imply that most of the $N_2O$ in ice wedges is extracted by three melting-refreezing cycles, such that only a small amount of $N_2O$ is left adsorbed or entrapped in ice-wedge soils. The authors posit that this might be attributed to the high solubility of $N_2O$ to water compared to $CH_4$.

P18, L375. «… easy to extract… »
➔ The typo is corrected.

P19, L397. « Our findings indicate that ... »
➔ Corrected.

**3) FIGURES AND TABLES**

Figure 1, P9.
- ➔ This is a pretty busy figure. We don't necessarily need the 3 legends (identical) in the middle. By removing them, more space could be created to enlarge the graphs a bit, because for now they are quite small. Also: what is the purpose of the insets (a-b, e-f)?
- ➔ The insets better show the data points in the low ranges. The modified version of Figure 1 is shown below.

[Figure]

a) I don't get the thing about the error bars (in blue). Are these 5x, 100x or 500x larger or smaller in 'real life' than displayed on the graphs? Not clear.

➔ The data uncertainties are too small to be plotted. Thus, the blue error bars in Figure 1 are magnified by 5x, 100, and 500x. We modify the figure caption for clarity. See our response below.

b) Explain what does 'hit5' mean.

➔ The meaning of 'hit5' and 'hit100' are already explained in the main text of Section 3.3. In addition, we add a sentence explaining 'hit5' in the figure caption for clarity.

➔ (P9.) **Figure 1.** Comparison of $CH_4$ and $N_2O$ mixing ratios and contents obtained by different extraction methods. Shown are scatter plots between wet- and dry (hit5) extraction results of $CH_4$ (a and b) and $N_2O$ (c and d), and between control- and biocide-treated wet extraction results for $CH_4$ (e) and $N_2O$ (f). The 'hit5' denotes the dry extraction with five times hitting (see Section 3.3). Left panels (a, c, and e) and (f) present in mixing ratios of gas in bubbles, while right (b) and (d) panels in moles of gas in a unit mass of ice (gas content). The sampling locations are indicated by different symbols. The color of each data point indicates the dry soil weight in the subsamples used in control wet extraction. The 1-sigma uncertainties of the mixing ratios (a, c, e, and f) are magnified by 5x, 20x, 100x, and 500x as denoted as blue error bars (see Appendix). The error bars are not visible where the error bars are smaller than markers. The grey dashed lines are 1:1 reference line. Note that the units of the axes of the insets in (e) and (f) are identical to the original plots. The p-value of two-sided Students' t-test of each comparison is denoted at the bottom right corner of each plot.

Table 1, P15.

a) It is not explained why the hit100/hit5 ratios for gas content (6th column) are much lower for most of the central Yakutia samples (Cyuie), compared to the other sites? This is indeed interesting, but why? Less soil aggregates in ice-wedge samples from this site, so relatively more bubbles and thus more extracted gas?

➔ We addressed this in the main text of Section 3.3. The low hit100/hit5 ratios of gas content in Cyuie samples are attributed to the easier crushing characteristics of the Cyuie samples compared with the others, so that much of the enclosed gases are extracted by hit5 extraction.

Figure 2, P18.

a) Where do the samples come from? CYC-02-B and CYC-03-C likely refer to Central Yakutia (Cyuie), but what about the other samples (C-04, C-30, C-10, C-12)? Please specify somewhere.

➔ The samples of 'C-##' come from Churapcha site. We modified the figure caption to specify the sample origin.

➔ (P18.) **Figure 2. Comparison of wet-extracted gas and residual gas for $CH_4$ and $N_2O$ mixing ratios (a and b) and contents (c and d).** The residual gas was extracted from the dry extraction method using the wet-degassed ice samples. The light green bars show the results of initial wet extraction, and the blue and red bars indicate the dry extraction of wet-degassed ice with 20- and 60-times hitting, respectively. The Cyuie samples are denoted as 'CYC', while 'C' indicates the Churapcha samples.

**Comments from the reviewer #2 and our responses:**

1) GENERAL COMMENTS

The paper reports novel aspects of experimental methods of gas extraction techniques. Currently, there are no works that conducted such a comparison, which has long been needed due to huge number of gas measurements in permafrost area conducted recently. Unfortunately, there is still no unified method for carrying out gas extraction that leads to the impossibility and impropriety of comparison the results obtained by different research groups.

The authors tested conventional wet and dry gas extraction methods for ice wedges coming to conclusion that current estimates of ground-ice gas budgets are likely underestimated. They found insignificant effects of microbial activity during wet extraction and significant difference in extraction results from polar ice cores and ice wedges. Therefore, the manuscript is of big interest for scientific community and contributes to changing our scientific understanding of a subject as it is has to be for TC.

The results are presented in well-structured way and the paper is easy to read.

However, there are a few general suggestions that could improve the article:
1)      I suggest adding a map of study sites, maybe some geological sections to get a better idea of the location and structure of ice wedges. Are they all Pleistocene?
➡ We agree with the reviewer and are including a map (see below) in our Supplement. The ages of the studied ice wedges have not been analysed, as they are beyond the scope of our manuscript.

[Figure]

**Supplementary Figure 1.** The site locations of the ground ice samples used in this study are marked in the map of circum-Arctic permafrost (Brown et al., 2002), yedoma distributions (Strauss et al., 2016), and major rivers.

2)      Besides it is really necessary to include the schemes of gas extraction procedures (both wet and dry techniques as well as the experiment on dry extraction efficiency and on residual gas contents after wet extraction.  Due to the limitation of the number of figures both 1 and 2 can be added as supplementary material.
➡ We add figures to the Supplement for clarity. As we already cited in the text, the details of the extraction system and procedures are well described in Ahn et al. (2009) and Shin (2014) for dry extraction and Yang et al. (2017) and Ryu et al. (2018) for wet extraction.

[Figure]

**Supplementary Figure 6.** Schematic diagram of the needle-crusher method together with enlarged photographs of crushing needles (left top), and extraction chamber (left bottom).

[Figure]

**Supplementary Figure 7.** Schematic diagram of melting-refreezing (wet extraction) procedure used in this study. More details about the wet extraction line and GC systems are described in Yang et al. (2017) and Ryu et al. (2018).

3)      Since the article is devoted to the comparison of methods, it would be useful to estimate the limits of applicability of the methods and measurement errors.

➔ We agree with the reviewer's comment, but the limits of applicability and measurement errors of both methods were already described in the main text (Section 3.3 and 3.4) and Appendix, respectively.

4)      It is necessary to add initial data on gas content and $CH_4$ and $N_2O$ mixing ratios as supplementary material to prove you main result about the same effectiveness of wet and dry extraction methods. As I see now from the Table 1 there can be 2 times difference (up to 20000 ppm) for $CH_4$

➔ We add our original data to our Supplement.

➔ We provide the relative amount of extracted gas in Table 1.

So the article is of big interest but needs major revision to be accepted and I would be happy to review a revision of the paper.

2) SPECIFIC COMMENTS AND EDITORIAL SUGGESTIONS

P.2 L50-51. «…ice sample was melted in a saturated sodium chloride (NaCl) solution, in order to minimize microbial activity and gas dissolution (Cherbunina et al., 2018 and references therein)». I found no mention in the article that NaCl was used in order to minimize microbial actvity, is it really there?

➔ We correct the reference as below:

➔ (P2. L49-52.) Other studies conducted by Russian scientists used an on-site melting method in which a large (1–3 kg) block of ground ice sample was melted in a saturated sodium chloride (NaCl) solution, in order to minimize  gas dissolution (Arkhangelov and Novgorodova, 1991).

➔ (P27. L488-490.) Arkhangelov, A. A., and Novgorodova, E. V.: Genesis of massive ice at 'Ice Mountains', Yenesei River, Western Siberia, according to results of gas analyses, Permafrost Periglac. Proc., 2, 167-170, http://doi.org/10.1002/ppp.3430020210, 1991.

P.2 L72. Please specify the size of the samples, add the site map and geological sections with sampling location

➔ A map of sampling sites is added as a Supplement Figure 1 (please refer to our response to General Comment #1 above). We also add images of sampling site outcrops.

[Figure]

**Supplementary Figure 2.** Images of ground ice outcrops at Churapcha (central Yakutia) site. Locations of the samples used in this study are indicated by yellow dotted circles.

[Figure]

**Supplementary Figure 3.** Images of ground ice outcrops at Cyuie (central Yakutia) sites: (a) ice wedge outcrop, (b) CYC and (c) CYB samples. Locations of the samples used in this study are indicated by yellow dotted lines.

[Figure]

**Supplementary Figure 4.** Images of ground ice outcrops at Zyryanka sites: (a and b) Zy-A, (c) Zy-B, and (d) Zy-F. Locations of the samples used in this study are indicated by yellow dotted lines.

[Figure]

**Supplementary Figure 5.** Images of ground ice outcrops at northern Alaskan sites: (a) Bluff03 and (b) Bluff06. Locations of the samples used in this study are indicated by yellow dotted boxes.

P.2 L104. Please include the schemes of gas extraction procedures
➔ Please refer to our response to General Comment #2.

P.2 L108. Why it is used precisely 5 times, not 100, can you reason it somehow?
➔ We have empirical knowledge that the polar ice core samples are well crushed within five times hitting during our dry extraction (needle-crusher) at SNU. For consistency of the analytical setup, we used an identical procedure for the dry extraction of ice-wedge samples to test whether the dry extraction method is applicable. In the meanwhile, tests with 100-times hitting were designed to understand the gas extraction efficiency and differences in gas mixing ratios for easily- and hardly crushed portions of ice wedges.

P.2 L144. It is not clear where did you get the dry soil mass before the extraction. «Taking the dry soil mass of the analysed samples (0.33 g) into account, we added 24 µL of saturated HgCl2 solution (at 20°C) to the sample flasks» Were there used the data on dry soil mass from the other samples? Because later you say « Dry soil content was measured using the leftover meltwater from the control-wet extraction tests. »
➔ We obtained the dry soil mass (0.33 g) from the leftover meltwater samples of the **previous** wet extractions, which was done to compare dry- and wet extractions. We revised the sentence for clarity.
➔ (P2. L145-150.) For biocide-treated tests, 1.84 mmol of mercuric chloride ($HgCl_2$) was applied per unit kilogram of soil, following established procedures for soil sterilization (Fletcher and Kaufman, 1980). We obtained the average dry soil mass (0.33 g) from the leftover meltwater samples of the previous wet extractions, which were carried out for comparison between dry- and wet extractions. Taking the average dry soil mass  (0.33 g) into account, we added 24 µL of saturated $HgCl_2$ solution (at 20°C) to the sample flasks.

P.12 L249. Please specify what do you mean by «ice hardness» here. As in L 249 «the extraction efficiency of the needle crusher not only depends on site characteristics, but also on the individual ice sample hardness», and later L. 251 «no relationship was observed between the dry soil content and the extraction efficiency», but L.274 « soil-rich ice has greater hardness than the soil-poor ice». I guess this is the matter of «soil aggregates» as you mention later, so the hardness in this case is defined by this parameter? Is it possible to quantify this?

➔ We observed that the samples with large-sized soil aggregates were difficult to crush with our needle-crusher system. Because no significant relationship was found between dry soil content and the extraction efficiency, the important parameter controlling the hardness may be the presence of large-sized aggregates, rather than just the soil content. Unfortunately,

> we do not have quantitative measurements of size (or volume) of each soil aggregate. Further study with three-dimensional image analysis will be useful to address this.

➔ To clarify this issue, we reword the sentence L.274 "soil-rich ice has greater hardness than the soil-poor ice" as follows:

➔ (P13. L276-280.) Thus, the hit5 $CH_4$ mixing ratios of the Cyuie samples may more reflect the gas mixing ratios in bubbles, while the hit100 results reflect more of the contribution from gas adsorbed on soil and trapped within soil aggregates than the hit5 results because the ice sample containing larger-sized aggregates has greater hardness than those with smaller aggregates or fine particles.

P.12 L255. Please specify the size range for «This is because the large-sized uncrushed soil aggregates or particles may have prohibited the needle crusher from crushing the small-sized ice flakes or grains». As the presence of the aggregates is one of the main limits to use the technique, is it possible to make at least a rough estimate of the amount of gas that can remain there?

➔ Although we have no quantitative measure of the size of soil aggregates, the aggregates in the studied ice wedges were observable by the naked eye as there was a clear contrast of darkness when back-lighted. Empirically, the size of observable aggregates ranged from millimetres to centimetres.

➔ However, it was not possible to estimate the amount of gas remaining in the uncrushed portion (mostly Zyryanka and Bluff samples), because the amount of entrapped gas is unknown. In contrast, for easily crushed samples (i.e. Cyuie samples), the hit100/hit5 ratio of gas content could be used to estimate the gas amount in soil aggregates. However, we cannot recommend this estimation because a certain portion of gas in soil aggregates could have been extracted by the hit5 procedures, and the amount of soil aggregates that are crushed or uncrushed after the hit5 procedures remains unknown.

P.12 L259. «Therefore, we do not recommend using a needle crusher system to measure gas contents in ice-wedge samples». Can you estimate the efficiency of the method in % in the same way it has been done for polar ice core ice samples (80–90%) (Shin, 2014)? As I see from the Table 1 the procedure «Hit5+Hit100» in most cases allows to extract more gas then the wet method even if uncrushed aggregates still occur. Can you recommend using dry extraction method in this modification?

➔ We cannot recommend this method. To measure the gas content precisely, near-perfect gas extraction from ice wedge samples is required. As the reviewer noted, the amount of gas extracted from both hit5 and hit100 procedures is generally higher than wet extraction. However, we hesitate to make a general statement because uncrushed soil aggregates may still exist even after hit100 extraction, depending on sampling locations and individual samples.

P.12 L272. Since you talk about gas in bubbles here : «the hit5 $CH_4$ mixing ratios of the Cyuie samples may more reflect the gas mixing ratios in bubbles, while the hit100 results reflect more of the contribution from gas adsorbed on soil and trapped within soil aggregates than the hit5 results» and further, may be it would be useful to get the data on ice porosity to compare with the results of extracted volume of gas since the volume of gas normalized to layer pressure approximately corresponds to porosity.

➔ This is a great idea. However, the relationship between the porosity and the normalized volume is applicable only when the gas extraction efficiency is close to 100%, or near constant. The gas extraction efficiency of the ice wedge is highly variable and difficult to measure precisely, limiting the estimation of the porosity.

P.16 L320. Please explain if I understand correctly the next paragraph:

«To examine how well the gas is extracted by wet extraction, we applied the dry extraction method to refrozen ice-wedge samples after wet extraction. We first prepared degassed ice-wedge samples that had undergone repetitive wet extractions (wet-degassed ice hereafter). Once the wet extraction experiments were completed, we repeated two cycles of melting-refreezing and evacuation procedures to degas the ice melt. After degassing by a total of three cycles of wet extraction and evacuation, the outermost surfaces (~2 mm) of the wet degassed ice were trimmed away in the walk-in freezer at SNU on the morning of experiments. The wet-degassed ice was then inserted into the needle crusher and the crusher chamber was evacuated. A specific amount of standard air was injected. Then, the wet-degassed ice samples were hit 20 or 60 times by the needle crusher.»

After the first freezing-melting cycle the sample gas in the headspace of the flask was collected and gas content, $CH_4$ and $N_2O$ ratio was measured. Then the two cycles were conducted. (and my question here is what happened to the gas in the flask-was it collected and measured or just evacuated) and next step was measuring the gas content in degassed ice through needle-crusher procedure.

➔ The reviewer has understood the text correctly. Regarding the question, we did not collect the gas extracted by second and third cycles of melting-refreezing procedures.

P.16 L328. Explain please why were such parameters chosen if in the previous dry extraction procedure you used 5 and 100 hits: « Then, the wet-degassed ice samples were hit 20 or 60 times by the needle crusher»

➔ The main goal of the tests with wet-degassed ice samples was to know if any gas remained after three cycles of wet extraction. We chose hitting 20 times instead of 5 times because significant amount of gas was already extracted during the three cycles of wet extractions. We also chose hitting 60 times to check if there is a significant difference in the amount of the extracted gas between hitting 20- and 60 times.

P.16 L331. The tests using the wet-degassed ice show an additional gas extraction of 43 to 88% of the amount of gas extracted during the initial wet extraction. I suggest to add this information to the conclusions as it is of big significance as well as if I get it right the best way of degassing the sample according to your manuscript is to combine three cycles of wet extraction with dry extraction for the residual gas. I think this has to be one of the main conclusion.

➔ The more number of wet and dry extractions, the better the gas extraction efficiency. We would prefer to leave the conclusions as is, since we cannot specify the best combination in numbers of the two extraction methods.

P.19 L391. Please specify what do you mean by «relatively soft ice wedges».

➔ Here we refer to the ice wedges that are more easily crushed than others by the hit5 procedure. To specify this, we revise the sentence as follows:

➔ (P19. L400-403.) In the meantime, we propose that both existing techniques may be suitable for gas mixing ratio measurements for bubbles in relatively soft ice wedges (i.e., easily crushed ice wedges by a hit5 extraction, e.g., Cyuie ice wedges in this study).

P.19 L 392  It seems to me that you have very good results of applying the method of three times wet extractions + residual gas extracted by a needle crusher for $N_2O$ and I don't get why there is in conclusions «Exceptionally, the N2O content in ice wedges may be measured by using repeated wet extractions, but this is not the case for determining the N2O mixing ratio»

➔ As the reviewer pointed out, our results indicate that repeated melting-refreezing procedures extract most of the $N_2O$ from ice wedges. However, we cannot guarantee the $N_2O$ mixing ratio because the relative extraction efficiency for each gas species may be variable. To clarify, we add below sentences in the main text.

➔ (P18. L374-376.) It should be noted that combination of repetitive wet extractions with dry extraction does not guarantee reliable estimation of $N_2O$ mixing ratio, because extraction efficiency of the other gas components may be different from that of $N_2O$.

3) FIGURES AND TABLES

Table 1. Add a column of dry soil content as in table 2

➔ We revise Table 1 as suggested (see next page).

The table with the data used for Fig.1 need to be added to get the difference between wet and dry method results.

➔ We add a table for the data used for Fig. 1 in the Supplement.

[revised manuscript text omitted]

---

## Referee Report (RR1)

**1) GENERAL COMMENTS**

The authors made a notable effort to deal with all the comments and suggestions I made in my first review, and I thank them very much for that.

Re. general comments #2 (site map, field and lab protocol pictures) and #4 (statements not supported by figures, data or references), the authors answered to the concerns I raised, and I have nothing more to add.

Re. general comment #1 (broad-audience impact), I still think that this work is relevant for a rather specialized audience, but I understand the points brought by the authors. I'll let the final decision to the Editor and I'm OK with the acceptance of the manuscript (after minor revision, see below).

Re. general comment #3 (main text structure, methods *vs*. results sentences), I totally get it that this paper mainly focuses on experimental/methodological aspects, as the authors say. In fact, the manuscript reads easily as is (as I already mentioned in my first review). Still, many sentences appearing in the Results and Discussion section are indeed associated to the work conducted in the lab, i.e. methods. If the Editor agrees to accept this non-conventional structure, I am OK with that.

I made a few (minor) specific comments, see below.

General question: What is the difference between 'Appendix' figures (included in the submitted manuscript) and 'Supplementary' figures (appearing in a separate file)? Why not group them all in the same category?

A final, important remark: according to the journal's guidelines, the data availability question (P26, L467-468, see below) should be settled BEFORE publication, not after.

**2) SPECIFIC COMMENTS AND EDITORIAL SUGGESTIONS**

P= page number, L = line number.

P1, L18-19. « However, existing gas extraction methods HAVE not been well tested. »

P2, L30-31. Carbon and nitrogen have just been mentioned in the line above, so I suggest using 'C' and 'N' from now on (« … temporarily removing this frozen C and N from active global cycles. »).

P4, L98. If we are talking about the gas mixing ratios of the ice-wedge ice, then we should read « … in that ITS gas mixing ratios are not homogenous… ».

P8, L190-193. I don't understand this sentence. Maybe it's related to the phrase 'of in centimetre scales'. What does it mean, this 'OF IN' phrase? Is there a word missing here, or a word that should not appear?

P11, L240-241. In my first review I suggested to display this information as a figure, rather than in a busy table (former manuscript version: P11, L236-237). The authors prefer to keep it as a table, arguing that the large ranges in CH4 and N2O mixing ratios make figures less readable. To me, the supplementary figure provided by the authors in their reply (same page, just below) looks great, and we see the significant differences not only between sites (central Yakutia, eastern Siberia, Alaska), but also between extraction techniques ('hit5' *vs*. 'hit100'). I thus prefer by far a figure, compared to a table. However, if the Editor also favors a table, then I would suggest to at least add some basic stats

(min-max-mean-SD) below each of the 3 sites, so that the reader can easily and efficiently have a 'big picture' view of these data.

P14, L311. « Even though a small  contamination does exist, … » (remove 'of').

P17, L357-359. Please add a reference (re. solubility of N2O in water compared to CH4).

P20, L409. Repetition of the bullet point just above (« Our findings indicate… »). Suggestion: « These results indicate/suggest that… ».

P26, L467-468. Data availability: How to make sure that the data will indeed be uploaded in a public repository (e.g., Pangaea) only AFTER publication? For several journals now, including TC, the datasets must be CITED within the text and included in the reference list (including an individual DOI): https://www.the-cryosphere.net/about/data_policy.html
I thus suggest submitting the datasets to a public repository with the FAIR approach ('findable, accessible, interoperable, and reusable'), and launch the process BEFORE the manuscript is accepted for publication. I let the Editor take the final decision about this.

**3) FIGURES AND TABLES**

See general comment above: some figures are in the Appendix, and some others (map, field and lab protocol pictures) appear separately in the Supplementary document. Why?

---

## Author Response (AR2)

**We would like to thank again the reviewers and the Editor for their careful review of our major revision. Our point-by-point responses to all the comments and annotations are presented below. The comments from the reviewer are shown in black, and our responses in red. Where indicated, additions to the manuscript are shown in blue and deleted text is indicated by red strikethrough, and the pages and line numbers are based on our revised (non-track) manuscript. The English was further improved by a professional editing company.**

=======================================================================

**Comments from the reviewer #1 and our responses:**

**1) GENERAL COMMENTS**

The authors made a notable effort to deal with all the comments and suggestions I made in my first review, and I thank them very much for that.

Re. general comments #2 (site map, field and lab protocol pictures) and #4 (statements not supportedby figures, data or references), the authors answered to the concerns I raised, and I have nothingmore to add.

➔ We thank the reviewer for his/her positive assessment of our revision.

Re. general comment #1 (broad-audience impact), I still think that this work is relevant for a rather specialized audience, but I understand the points brought by the authors. I'll let the final decision to the Editor and I'm OK with the acceptance of the manuscript (after minor revision, see below).

➔ As addressed in our major revision, we believe that our findings are important in broad community working on permafrost as well as climate sciences. We thank the reviewer for his/her understanding.

Re. general comment #3 (main text structure, methods vs. results sentences), I totally get it that thispaper mainly focuses on experimental/methodological aspects, as the authors say. In fact, themanuscript reads easily as is (as I already mentioned in my first review). Still, many sentencesappearing in the Results and Discussion section are indeed associated to the work conducted in thelab, i.e. methods. If the Editor agrees to accept this non-conventional structure, I am OK with that.

➔ We agree that the conventional structure is an effective way to organize scientific articles. However, each experiment has own rationale and our manuscript was written following the logical flow, which would be difficult to maintain in the conventional format. Thus, we believe that current structure is best for our manuscript.

I made a few (minor) specific comments, see below.

General question: What is the difference between 'Appendix' figures (included in the submittedmanuscript) and 'Supplementary' figures (appearing in a separate file)? Why not group them all in the same category?

➔ We prefer to put the methodological aspects (i.e., correction and uncertainty estimate) and the results of complementary experiments in Appendix rather than Supplement for readers to find them easily. Instead, we put the maps, pictures, and schematic diagrams in Supplement as a separate file to reduce the total length of main text, but the readers are still accessible if they are interested. We believe this criteria is not unreasonable, but certainly we will follow the Editor's decision.

A final, important remark: according to the journal's guidelines, the data availability question (P26,L467-468, see below) should be settled BEFORE publication, not after.

➔ We submitted our dataset to Zenodo data repository, and added citation following the journal's policy.

**2) SPECIFIC COMMENTS AND EDITORIAL SUGGESTIONS**

P= page number, L = line number.

P1, L18-19. « However, existing gas extraction methods HAVE not been well tested. »

➔ Done.
➔ However, existing gas extraction methods have not been well tested.

P2, L30-31. Carbon and nitrogen have just been mentioned in the line above, so I suggest using 'C'and 'N' from now on (« … temporarily removing this frozen C and N from active global cycles. »).

➔ Revised as suggested.
➔ Permafrost preserves large amounts of soil carbon (C) and nitrogen (N) in a frozen state (e.g., Hugelius et al., 2014; Salmon et al., 2018), temporarily removing this frozen C and N from active global cycles.

P4, L98. If we are talking about the gas mixing ratios of the ice-wedge ice, then we should read « … inthat ITS gas mixing ratios are not homogenous… ».

➔ Done.
➔ Ice-wedge ice is  different from polar ice cores in that its gas mixing ratios are not homogeneous (e.g., Kim et al., 2019), which may hinder exact comparison with results from adjacent ice samples.

P8, L190-193. I don't understand this sentence. Maybe it's related to the phrase 'of in centimeter scales'. What does it mean, this 'OF IN' phrase? Is there a word missing here, or a word that should not appear?

➔ We revised the phrase as follows.
➔ We note that the heterogeneous distribution of gas mixing ratios at the centimetre scale (Kim et al., 2019) may not have been completely smoothed out by our sub-sample selection, although we randomly chose 8–12 ice cubes for each measurement.

P11, L240-241. In my first review I suggested to display this information as a figure, rather than in abusy table (former manuscript version: P11, L236-237). The authors prefer to keep it as a table,arguing that the large ranges in CH4 and N2O mixing ratios make figures less readable. To me, the supplementary figure provided by the authors in their reply (same page, just below) looks great, andwe see the significant differences not only between sites (central Yakutia, eastern Siberia, Alaska), but also between extraction techniques ('hit5' vs. 'hit100'). I thus prefer by far a figure, compared to atable. However, if the Editor also favors a table, then I would suggest to at least add some basic stats (min-max-mean-SD) below each of the 3 sites, so that the reader can easily and efficiently have a 'bigpicture' view of these data.

➔ We replaced Table 1 with the figure we showed in our previous response, and the figure caption has been changed accordingly. Instead, we moved the original Table 1 to Appendix because our main text still refers to the Table, for example, the hit100/hit5 ratios.

P14, L311. « Even though a small of contamination does exist, … » (remove 'of').

➔ Done.

➔ Even though minor contamination did occur, its effects  had already been subtracted via blank correction and taken into account in the overall error estimation (see Appendix).

P17, L357-359. Please add a reference (re. solubility of N2O in water compared to CH4).

➔ We added below citation in the sentence as well as in the reference list.

➔ Fogg, P. G. T., and Sangster, J.: Chemicals in the Atmosphere: Solubility, Sources and Reactivity, John Wiley & Sons, Inc., 2003.

P20, L409. Repetition of the bullet point just above (« Our findings indicate… »). Suggestion: « These results indicate/suggest that… ».

➔ We agree with the reviewer, however, we revised the sentence as below because the previous phrasing had no add value and therefore was unnecessary. As this sentence is already a part of the conclusion section where we are already summarizing our results.

➔ Saturated NaCl solution is unnecessary for preventing microbial activity during melting, as employed by, e.g., Cherbunina et al. (2018).

P26, L467-468. Data availability: How to make sure that the data will indeed be uploaded in a publicrepository (e.g., Pangaea) only AFTER publication? For several journals now, including TC, thedatasets must be CITED within the text and included in the reference list (including an individual DOI):https://www.the-cryosphere.net/about/data_policy.htmlI thus suggest submitting the datasets to a public repository with the FAIR approach ('findable,accessible, interoperable, and reusable'), and launch the process BEFORE the manuscript isaccepted for publication. I let the Editor take the final decision about this.

➔ Please refer to our response above.

**3) FIGURES AND TABLES**

See general comment above: some figures are in the Appendix, and some others (map, field and labprotocol pictures) appear separately in the Supplementary document. Why?

➔ Please find our response to general comment.

[revised manuscript text omitted]